# Priming of lineage-specifying genes by Bcl11b is required for lineage choice in post-selection thymocytes

Satoshi Kojo[1], Hirokazu Tanaka[1], Takaho A. Endo [2], Sawako Muroi[1], Ye Liu[3], Wooseok Seo[1], Mari Tenno[1], Kiyokazu Kakugawa[1], Yoshinori Naoe[1], Krutula Nair[1], Kazuyo Moro[4], Yoshinori Katsuragi[5], Akinori Kanai[6], Toshiya Inaba[6], Takeshi Egawa[7], Byrappa Venkatesh[8], Aki Minoda[3], Ryo Kominami[5] & Ichiro Taniuchi[1]

T-lineage committed precursor thymocytes are screened by a fate-determination process mediated via T cell receptor (TCR) signals for differentiation into distinct lineages. However, it remains unclear whether any antecedent event is required to couple TCR signals with the transcriptional program governing lineage decisions. Here we show that Bcl11b, known as a T-lineage commitment factor, is essential for proper expression of ThPOK and Runx3, central regulators for the CD4-helper/CD8-cytotoxic lineage choice. Loss of Bcl11b results in random expression of these factors and, thereby, lineage scrambling that is disconnected from TCR restriction by MHC. Initial *Thpok* repression by Bcl11b prior to the pre-selection stage is independent of a known silencer for *Thpok*, and requires the last zinc-finger motif in Bcl11b protein, which by contrast is dispensable for T-lineage commitment. Collectively, our findings shed new light on the function of Bcl11b in priming lineage-specifying genes to integrate TCR signals into subsequent transcriptional regulatory mechanisms.

[1] Laboratory for Transcriptional Regulation, RIKEN Center for Integrative Medical Sciences (IMS), 1-7-22 Suehiro-cho, Tsurumi-ku, Yokohama 230-0045, Japan. [2] Laboratory for Integrative Genomics, RIKEN Center for Integrative Medical Sciences (IMS), 1-7-22 Suehiro-cho, Tsurumi-ku, Yokohama 230-0045, Japan. [3] Division of Genomic Technologies, RIKEN Center for Life Science Technologies (CLST), 1-7-22 Suehiro-cho, Tsurumi-ku, Yokohama 230-0045, Japan. [4] Laboratory for Innate Immune Systems, RIKEN Center for Integrative Medical Sciences (IMS), 1-7-22 Suehiro-cho, Tsurumi-ku, Yokohama 230-0045, Japan. [5] Division of Molecular Biology, Department of Molecular Genetics, Graduate School of Medical and Dental Sciences, Niigata University, Niigata 951-8510, Japan. [6] Department of Molecular Oncology, Research Institute for Radiation Biology and Medicine, Hiroshima University, 1-2-3, Kasumi, Minami-ku, Hiroshima 734-8553, Japan. [7] Department of Pathology and Immunology, School of Medicine, Washington University School of Medicine, 660 S Euclid, Saint Louis 63110 MO, USA. [8] Institute of Molecular and Cell Biology, Agency for Science, Technology and Research, Biopolis 138673, Singapore. Correspondence and requests for materials should be addressed to I.T. (email: ichiro.taniuchi@riken.jp)

Early thymocyte progenitors (ETPs) retain their developmental potential to become non-T- lymphoid cells, but, upon exposure to the thymic microenvironment, the ETP expression program is dramatically altered to commit them to the T-lymphoid lineage[1, 2]. Previous studies demonstrated that a final T-lymphoid lineage commitment occurs at a developmental checkpoint during the transition from the CD4−CD8− double negative (DN)2a to the DN2b stage, where expression of the transcription factor Bcl11b is induced[3, 4]. Bcl11b-deficiency results in a developmental arrest of early T cell progenitors at the DN2a stage along with a concomitant acquisition of myeloid-lineage/natural killer (NK) cell gene signatures[5]. Bcl11b thus serves as a T-lineage commitment factor that eliminates developmental potential to non-T lymphoid cells at the DN2 checkpoint[3–5]. Beside its function during early thymocyte development, Bcl11b continues to be expressed during T lymphocyte differentiation[5, 6] and modulates the development of many lymphoid subsets, including natural killer T (NKT)[7], regulatory T (Treg)[8] and type2 innate lymphoid (ILC2) cells[9–11].

After commitment to the T-lineage, another checkpoint, known as β-selection, selectively expands DN3 cells that successfully express functional TCRβ chain after V(D)J recombination of the *Tcrb* locus. Thymocytes that have passed β-selection become αβT-lineage-restricted CD4+CD8+ double positive (DP) precursor thymocytes, which express complete αβTCR complexes for the first time during T cell development. DP precursors are then subjected to additional positive and negative selections that enrich precursors with TCRs recognizing

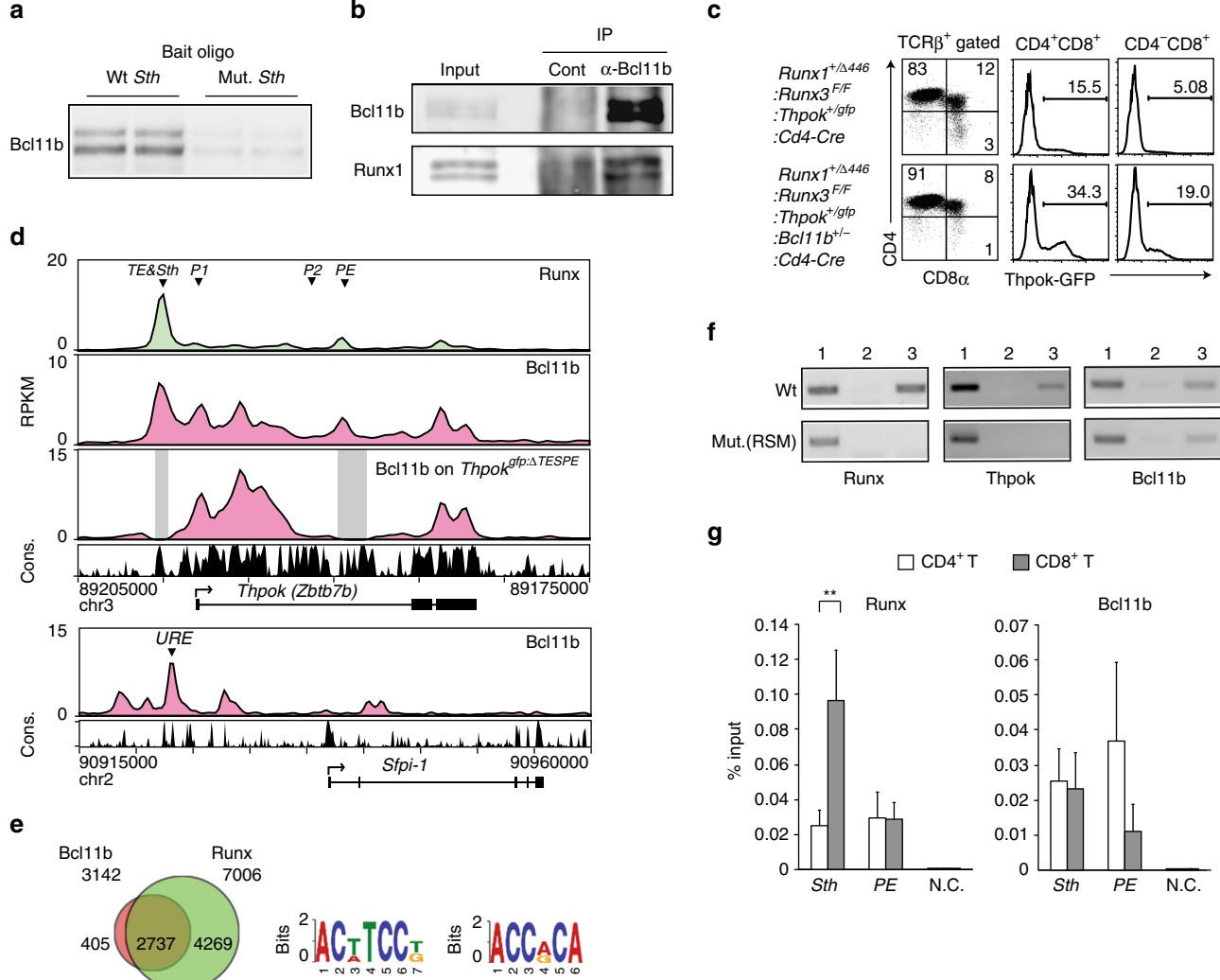

**Fig. 1** Binding of Bcl11b to regulatory regions in the *Thpok* gene. **a** DNA pull-down assay showing in vitro Bcl11b binding to wild-type (Wt) *Thpok* silencer (*Sth*) core sequences, but not efficiently to mutant (Mut.) sequences. One representative of two experiments. **b** Co-immune precipitation assay showing interaction of Bcl11b with Runx1. One representative of two experiments. **c** Flow cytometry showing an increase of CD8+ T cells de-repressing Thpok-GFP upon reduction of *Bcl11b* dosage in *Runx* mutant mice. One representative of two independent experiments. **d** Bcl11b ChIP-seq tracks at the *Thpok*, *Thpok*$^{gfp:ΔTESPE}$, and *Sfpi1* genes in total thymocytes along with Runx ChIP-seq (GSE90794) tracks at the *Thpok* gene (*top*) for reference. Gene structure, transcriptional orientation and conservations in mammals (Cons.) in the *Thpok* gene are indicated. Positions of *Thpok* silencer (*Sth*), two enhancers (*TE* and *PE*), distal P1-promoter (*P1*) and proximal enhancer (*PE*) in the *Thpok* gene, and upstream regulatory element (*URE*) in the *Sfpi1* gene are indicated as *arrowheads*. **e** Co-occupancy by Runx at Bcl11b-bound genome regions. Runx recognition site (5′-ACCPuCA-3′) was listed among the top two common sequences for Bcl11b-bound regions. **f** ChIP-PCR assay for binding of Runx, Bcl11b, and ThPOK to Wt and Runx site mutated (RSM) *Sth* regions in CD4+ T cells from *Thpok*$^{+/gfp:241-401RM}$ mice. Lanes 1, 2, and 3 are input, control IgG and antibody-against the indicated transcription factor, respectively. One representative of two independent experiments. **g** ChIP-qPCR analyses showing binding of Runx and Bcl11b to the *Sth* and *PE* in CD4+ (white bars) and CD8+ (gray bars) T cells. Combined data from three independent ChIP experiments is shown. **P < 0.01 (unpaired *t*-test)

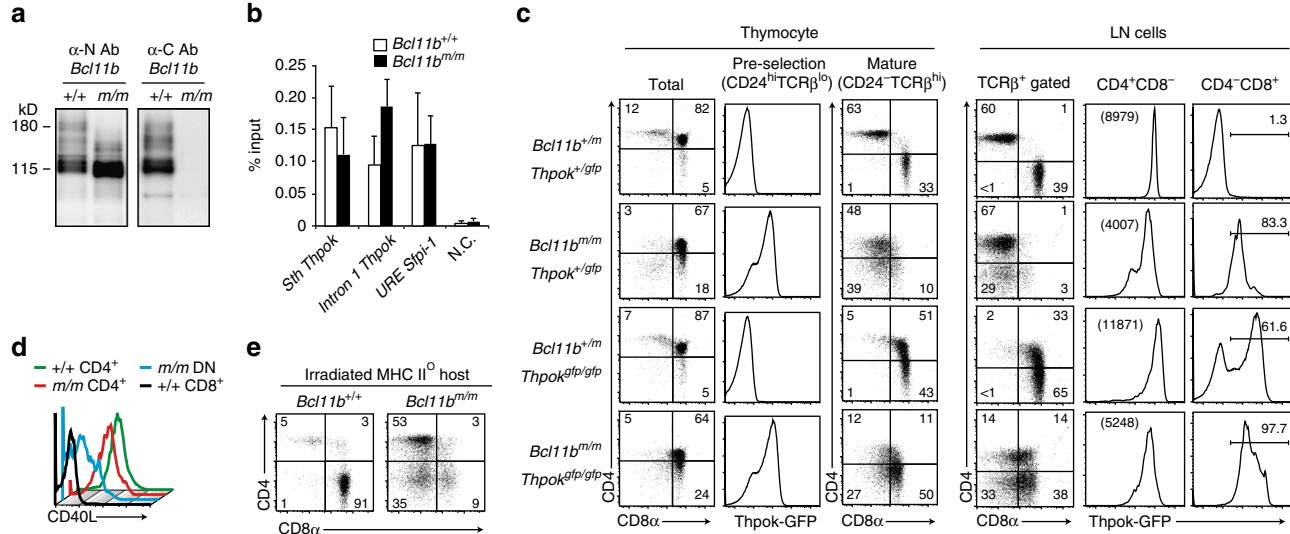

**Fig. 2** CD4-skewed development by impaired *Thpok* regulation from hypomorphic *Bcl11b*$^{m/m}$ progenitors. **a** Expression of truncated Bcl11b protein in the neonatal thymus from the hypomorphic *Bcl11b*$^m$ allele detected by an antibody recognizing the N- (*left*), but not the C- (*right*), terminus of Bcl11b. One representative of two experiments. **b** Summary of ChIP assay for binding of Wt (white bars) and hypomorphic (black bars) Bcl11b to indicated regions in total neonatal thymocytes. **c–e** Flow cytometry analyzing the expression of CD4, CD8, and Thpok-GFP during T cell development in Rag1-deficient hosts **c** and irradiated MHC-II deficient hosts **e** reconstituted with fetal liver cells with the indicated genotypes. Data are representative of at least two independent experiments. Numbers in *dot plots* indicate the percentage of cells in each quadrant. Histograms showing CD40L expression 2 days after in vitro TCR stimulation of peripheral CD4$^+$, CD8$^+$ and CD4$^-$CD8$^-$ DN cells differentiated in Rag1-deficient mice from *Bcl11b*$^{+/+}$ and *Bcl11b*$^{m/m}$ fetal liver cells **d**. Data are representative of two independent experiments

antigen-MHC complexes with intermediate affinity but eliminate precursors expressing self-reactive TCRs, respectively[12]. Positively selected thymocytes differentiate into distinct T cell subsets with distinct functionalities via the activation of specific developmental programs. For instance, positive selection signaled via TCR engagement by MHC class-II (MHC-II) and class-I (MHC-I) guides precursors to differentiate into CD4$^+$CD8$^-$ helper or CD4$^-$CD8$^+$ cytotoxic T cells through the induction of key transcription factors, ThPOK and Runx3, respectively[13, 14]. Thus, DP precursors must be ready to integrate TCR signals, translating them into the appropriate developmental program. However, an important gap in our understanding of these processes is how TCR signals are coupled to mechanisms that control the expression of lineage-specifying genes, and it remains unclear whether preceding events are required for this coupling.

One such lineage-specifying transcription factor, Zbtb7b, also known as T-helper-inducing POZ/Krueppel$^-$ like factor (ThPOK), is a member of the BTB-POZ zinc-finger transcription factor family[15] and is encoded by *Zbtb7b*, hereafter referred to as the *Thpok* gene. Previous genetic studies for 'gain and loss' of ThPOK function demonstrated that its presence or absence in post-selection thymocytes is a major determinant of the CD4-helper (ThPOK$^+$) versus CD8-cytotoxic (ThPOK$^-$) lineage dichotomy[16–18]. Consistent with these findings, expression of the *Thpok* gene is restricted to MHC-II selected thymocytes[16] in positively selected thymocytes. Accordingly, *Thpok* regulation has been recognized as an ideal model to study how TCR signals are coupled with the transcriptional program that establishes the identity of CD4$^+$ helper T cells. Such studies identified a transcriptional silencer in *Thpok*, hereafter denoted as *Thpok* silencer (*Sth*), which is an essential cis-acting element restricting *Thpok* expression to post-selection thymocytes in the helper lineage[19, 20]. In addition to *Sth*, there are at least three enhancers in the *Thpok* locus[20]. Among them, the thymic enhancer (*TE*), located upstream of the *Sth*, acts first to initiate *Thpok* expression[21], which is subsequently upregulated through the activity of a

proximal enhancer (*PE*) locating 1.8 kb downstream of the proximal P2-promoter[22]. While factors that regulate *Sth* activity, such as Runx family proteins[19] and MAZR[23, 24], have been identified, the factors involved in the activation of *TE* and *PE* remain poorly characterized. Gata3[25] and Tcf1/Lef1[26] regulate *Thpok* expression, but primarily do so by targeting other regulatory regions such as general T-lymphoid enhancer, the known third enhancer. In contrast to *Thpok* regulation, very little is known about transcription factors that orchestrate CD8$^+$ T cell-specific expression of *Runx3* from its distal P1-promoter[27]. Signals emanating from the IL-7 cytokine receptor have been shown to activate *Runx3*[28, 29]; however, the intermediaries and their cis-regulatory targets in *Runx3* remain to be established[30].

Here we report two mechanisms by which Bcl11b governs the transcriptional program dissecting helper versus cytotoxic lineage commitment: *Sth*-independent repression of *Thpok* in DN thymocytes and enhancer-dependent *Runx3* repression in CD4-lineage cells. Deletion of Bcl11b in thymocytes at post-β-selection stage causes chaotic *Thpok* and *Runx3* expression, inducing random differentiation of both MHC-I and MHC-II selected cells into the helper and cytotoxic subsets. Along with earlier requirement for Bcl11b prior to DP stage in later *Foxp3* activation, we conclude that lineage-specifying genes are 'primed' by Bcl11b before or during transition to the DP stage to represent an essential event for coupling TCR signals to expression programs for differentiating into the appropriate T-effector subsets.

## Results

**Bcl11b binds to the *Thpok* locus.** Based on our assumption that proteins bound to *Thpok* silencer (*Sth*) should mediate coupling of TCR signals during positive selection with release of *Thpok* expression, we attempted to purify protein complexes by in vitro capture with an oligo-nucleotide harboring core *Sth* sequences. Consistent with prior chromatin immunoprecipitation (ChIP) assays[31], Bcl11b protein was efficiently enriched by affinity

purification with core *Sth* sequences, but to lesser extent with mutant *Sth* sequences (Fig. 1a). Bcl11b also associated with other known *Sth* binding proteins, Runx1 (Fig. 1b). Furthermore, reduction of *Bcl11b* dosage to the half (*Bcl11b*$^{+/-}$) in the *Runx* mutant mice, in which combined mutations of *Runx1* and *Runx3* genes attenuated *Sth*-mediated *Thpok* repression[32], resulted in an increase of CD8$^+$ subset de-repressing a *Thpok*$^{gfp}$ reporter[22] (Fig. 1c), indicating genetic interaction between two factors in the

regulation of *Sth* function. Our ChIP sequencing (ChIP-seq) assay of total thymocytes also detected that Bcl11b associated with *Sth* and proximal enhancer (*PE*) as well as intronic regions downstream from distal P1-promoter in the *Thpok* locus (Fig. 1d). Bcl11b also bound to an upstream regulatory element (URE)[33] in the *Sfpi-1* locus, which encodes a myeloid transcription factor, PU.1, and is a putative target of Bcl11b to eliminate myeloid potential during T-lineage commitment (Fig. 1d). A more

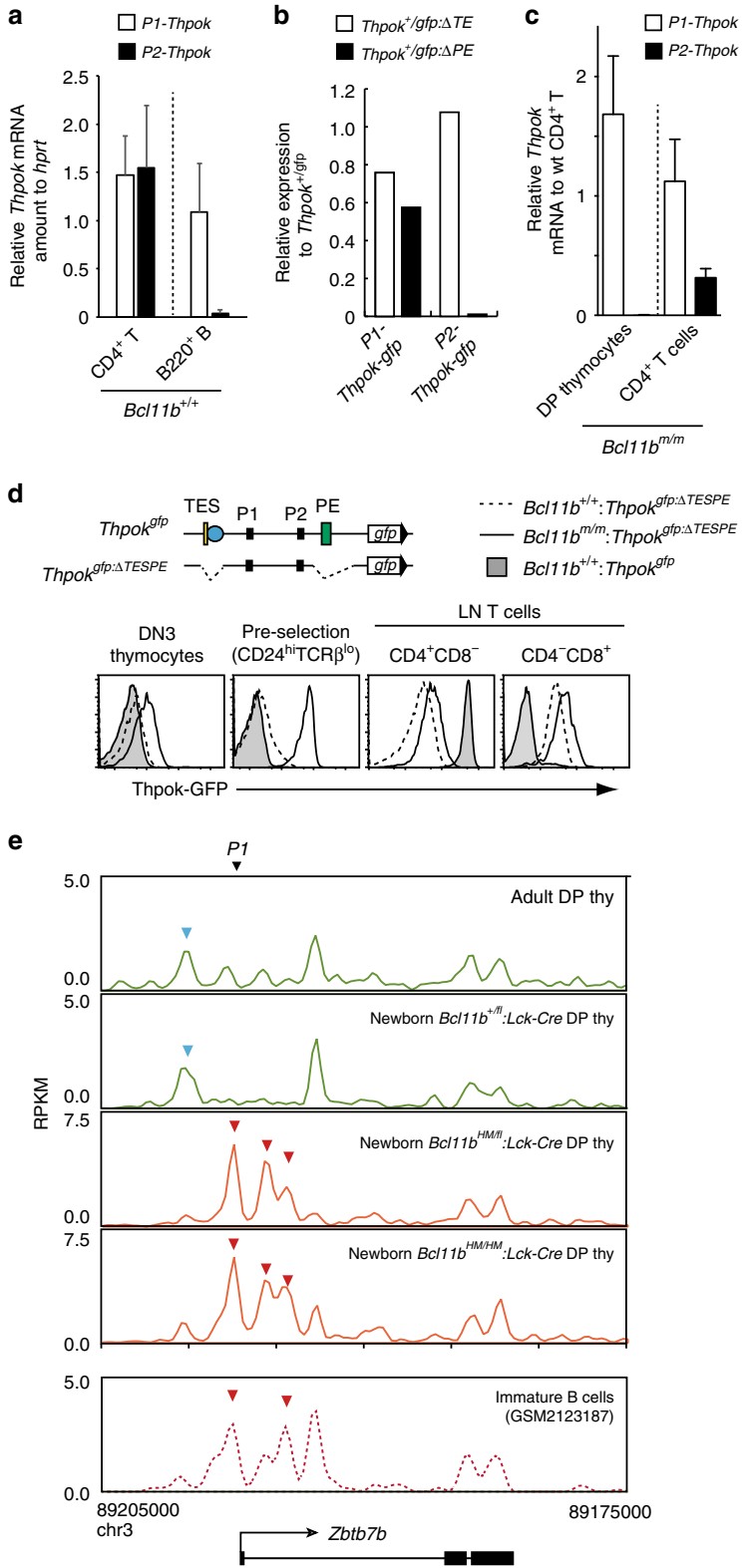

global analysis revealed the Runx recognition motif (5′-ACCPuCA-3′) as the second-ranked sequence enriched at Bcl11b-bound regions in thymocytes (Fig. 1e) and Runx bindings coincided with ~85 % of Bcl11b-bound regions (Fig. 1e), which is consistent with previous observation using neuronal cells[34]. However, Bcl11b binding to *Sth* was Runx-independent, as determined using primary CD4[+] T cells in which two Runx sites within the *Sth* were mutated by a knock-in on the *Thpok*[gfp] reporter allele[35], whereas ThPOK binding to the *Thpok* silencer requires Runx-binding (Fig. 1f). Our analytical ChIP assay in CD4[+] and CD8[+] T cell subset detected that Bcl11b binding at *PE* was greater in Thpok-expressing CD4[+] T cells, while fivefold more Runx was bound to *Sth* in ThPOK-negative cells (Fig. 1g). These results suggest that Bcl11b directly regulates activity of *Thpok* locus through binding to the silencer and enhancer elements.

**Impaired *Thpok* regulation in *Bcl11b* mutant mice.** Because there is a developmental arrest at the DN2a stage when Bcl11b function is completely lacking, it is necessary to utilize a *Bcl11b*[fl] allele to examine the function of Bcl11b during T cell development beyond the DN stage. We therefore attempted to generate our own *Bcl11b*[fl] allele. However, our first trial unexpectedly generated a mutant *Bcl11b* allele, hereafter referred as to a *Bcl11b*[m], which produced a truncated Bcl11b protein due to a frameshift mutation incorporated into the targeting vector during its construction (Fig. 2a and Supplementary Fig. 1a–c). Despite its loss of the final zinc-finger motif, truncated mutant Bcl11b protein could bind to the *Sth* and an intronic region in the *Thpok*, as well as to the URE in *Sfpi-1* (Fig. 2b). Similar to *Bcl11b* null animals (*Bcl11b*[−/−]), homozygous *Bcl11b*[m] mice died as neonates (Supplementary Fig. 1d). However, in contrast to the null animals, which completely lack CD4[+]CD8[+] DP thymocytes and thereby have small thymus[36], we noticed that total thymocyte number is only slightly reduced in *Bcl11b*[m/m] neonates (Supplementary Fig. 1e) and, more strikingly, the DP population was present in those mice (Supplementary Fig. 1f). In addition, *Bcl11b*[m/m] progenitors gave rise to ILC2 cells (Supplementary Fig. 1g), a subset that does not develop from *Bcl11b*[−/−] progenitors[9, 10]. These results indicated that the mutant *Bcl11b*[m] is a hypomorphic allele, thereby hereafter its product is referred to as hypomorphic Bcl11b protein (Bcl11b[HM]).

We next investigated the T cell development of *Bcl11b*[m/m] progenitors that also harbor a *Thpok*[gfp] reporter allele. For this purpose, we transferred *Bcl11b*[m/m]:*Thpok*[+/gfp] or control *Thpok*[+/gfp] fetal liver progenitors into T cell-deficient *Rag1*[−/−] hosts, allowing differentiation of the progenitors to proceed for two months. As shown in Fig. 2c, Thpok-GFP expression was absent in pre-selection thymocytes (defined as CD24[hi]TCRβ[lo]) from control cells. In sharp contrast, nearly all pre-selection thymocytes derived from the transferred *Bcl11b*[m/m] progenitors

expressed Thpok-GFP. While the two major T cell subsets, CD4[+] helper and CD8[+] cytotoxic, were differentiated from control *Bcl11b*[+/+] progenitors, *Bcl11b*[m/m] progenitors gave rise to predominantly CD4[+] T cells with a helper-related signature (e.g., CD40L induction) in mature thymocyte and peripheral T cell populations (Fig. 2c, d). To address whether a re-direction of MHC-I selected cells to the CD4[+] lineage contributed to this CD4-skewing, we generated chimeras using irradiated MHC-II deficient hosts expressing the Ly9.1 marker, in which only differentiation of MHC-I selected cells was supported. Strikingly, we continued to observe skewing of CD4 differentiation in Ly9.1[−] donor-derived populations from *Bcl11b*[m/m], but not *Bcl11b*[+/+], progenitors (Fig. 2e). The shift to CD4 dominant differentiation was not observed when *Bcl11b*[m/m]:*Thpok*[gfp/gfp] progenitors lacking ThPOK expression were transferred (Fig. 2c). Along with de-repression of Thpok-GFP in remaining CD8[+] T cells (Fig. 2c), we conclude that aberrant *Thpok* expression redirects MHC class I-selected *Bcl11b*[m/m] thymocytes to the CD4[+] lineage. After these initial characterizations of the hypomorphic *Bcll1b*[m] allele, we generated a true conditional mutant (*Bcl11b*[fl]) allele and a mutant allele that retains the identical frameshift mutation to the *Bcl11b*[m] allele but lacks the loxP sequence in its 3′UTR, which we referred to as the *Bcl11b*[HM] allele (Supplementary Fig. 1a).

**Silencer-independent *Thpok* repression by Bcl11b.** To further investigate how *Thpok* expression was de-regulated in *Bcl11b*[m/m] cells, we examined promoter usages. Prior studies have shown that *Thpok* is transcribed from distal P1- and proximal P2-promoters[20], however, P2-derived transcripts were detected specifically in T-lineage cells and require *PE* activity (Fig. 3a, b). Given the loss of Thpok-GFP expression in peripheral T cells upon removal of both *TE* and *PE* (Supplementary Fig. 2a), *TE* is likely to drive P1-promoter activity in T cells. In contrast, *Thpok* transcript in *Bcl11b*[m/m] pre-selection thymocytes was transcribed only from the P1-promoter (Fig. 3c). To understand the underlying mechanisms for unusual P1-promoter activation without P2-promoter activation in *Bcl11b*[m/m] thymocytes, we examined Thpok-GFP expression from the *Thpok*[gfp:ΔTESPE] reporter allele, in which all three T cell-specific regulatory elements, *Sth*, *TE*, and *PE*, were removed from the *Thpok*[gfp] allele by sequential gene targeting (Fig. 3d). In Bcl11b wild-type cells, *Thpok* expression from the enhancer-deficient *Thpok*[gfp:ΔTESPE] reporter allele was derived solely from the P1-promoter (Supplementary Fig. 2b), and was detected only in cells after the post-selection stages (Fig. 3d). In contrast, despite deletion of the *Sth* silencer, Thpok-GFP expression from the *Thpok*[gfp:ΔTESPE] reporter allele was initiated in *Bcl11b*[m/m] thymocytes at DN3 stage and its expression level was increased during transition to DP stage (Fig. 3d). These data revealed the presence of undefined *Sth*-independent mechanism by which Bcl11b represses *Thpok* expression in DN3 as well as in DP thymocytes.

**Fig. 3** *Sth*-independent *Thpok* repression by Bcl11b in pre-selection DP thymocytes. **a** Quantitative RT-PCR showing amount of promoter-specific *Thpok* transcripts in splenic CD4[+] T cells and splenic B220[+] B lymphocytes. Summary of three measurements. **b** Quantitative RT-PCR for *Thpok-gfp* expression in CD4[+] T cells from mutant reporter alleles, *Thpok*[gfp:ΔTE] and *Thpok*[gfp:ΔPE], lacking the thymic enhancer (*TE*) or proximal enhancer (*PE*), respectively. Levels of P1- and P2 promoter-specific transcripts relative to those from the control *Thpok*[gfp] allele harboring intact enhancers is shown. One representative of two independent experiments. **c** Promoter-specific *Thpok* transcripts in *Bcl11b*[m/m] pre-selection DP thymocytes and CD4[+] T cells relative to those in control CD4[+] T cells. Summary of three measurements. **d** Histograms showing Thpok-GFP expression from the *Thpok*[gfp:ΔTESPE] allele in the indicated cell subsets differentiated in Rag1-deficient recipients from control (*dotted line*) and *Bcl11b*[m/m] (*solid line*) fetal liver cells. Expression pattern of *Thpok*[gfp] is shown as reference (*shaded*). Schematic structure of the *Thpok*[gfp:ΔTESPE] allele is included (*upper panel*). One representative of two independent experiments. **e** ATAC-seq tracks at the *Thpok* locus in CD24[hi]TCRβ[lo]CD4[+]CD8[+] pre-selection thymocytes of 6-week-old *Bcl11b*[+/+] mice, newborn *Bcl11b*[+/fl]:*Lck-cre*, newborn *Bcl11b*[HM/fl]:*Lck-cre* and newborn *Bcl11b*[HM/HM]:*Lck-cre* mice. ATAC-seq results in immature B cells (GSM2123187) is shown as reference. DP thymocyte- and B lymphocyte-specific peaks are marked with *blue* and *red arrowheads*, respectively. Position of the P1 promoter is indicated with *black arrowhead*

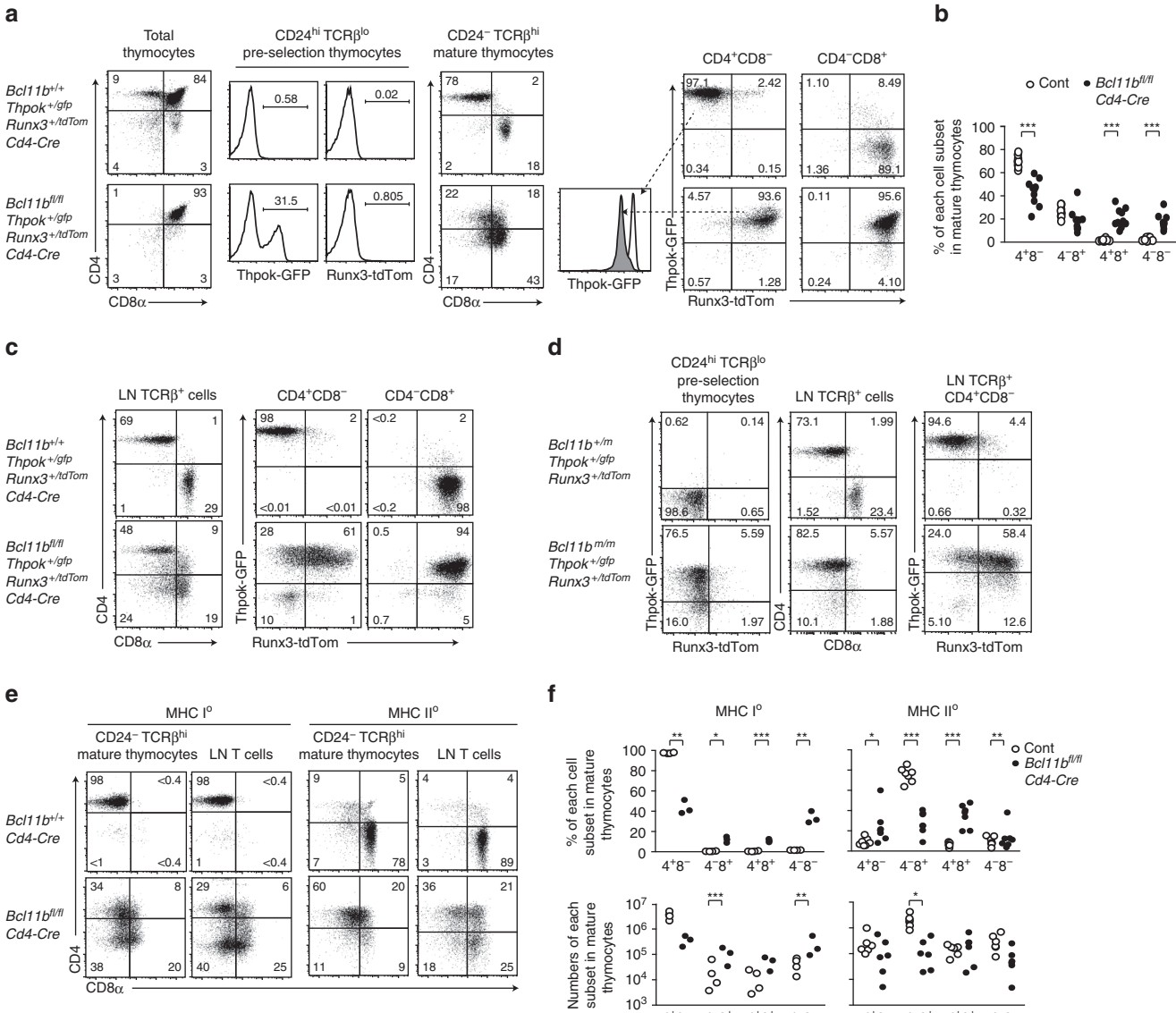

**Fig. 4** Lineage scrambling resulting from inactivation of *Bcl11b* at the DP stage due to sporadic *Thpok* and *Runx3* expression. **a** Flow cytometry analyzing the expression of CD4, CD8, Thpok-GFP, and Runx3-tdTomato by various thymocyte subsets in mice with the indicated genotypes. The histogram in the middle showing Thpok-GFP expression in CD4⁺CD8⁻ mature thymocytes from control (*open*) and *Bcl11b*-deficient (*shaded*) cells. **b** Graph showing summary of percentage of each subset in mature (CD24⁻TCRβʰⁱ) thymocytes population of mice with indicated genotype. **c** Flow cytometry analyzing the expression of CD4, CD8, Thpok-GFP and Runx3-tdTomato in lymph node T cells of mice with the indicated genotypes. One representative of at least independent mice. **d** Flow cytometry analyzing the expression of CD4, CD8, Thpok-GFP and Runx3-tdTomato in pre-selection DP thymocytes and lymph node T cells differentiated in Rag1-KO hosts from *Bcl11b*⁺/⁺ and *Bcl11b*ᵐ/ᵐ fetal livers harboring *Thpok*ᵍᶠᵖ and *Runx3*ᵗᵈᵀᵒᵐᵃᵗᵒ reporter alleles. One representative of two independent experiments. **e**, **f** Representative flow cytometry **e** analyzing differentiation of MHC-I and MHC-II selected cells in MHC-II and MHC-I deficient (MHC-II⁰ and MHC-I⁰) backgrounds, respectively, along with a graph **f** showing a statistical summary of the percentages and absolute numbers in each cell subset in CD24⁻TCRβʰⁱ mature thymocytes population of control (white circles) and *Bcl11b*ᶠˡ/ᶠˡ:*Cd4-Cre* (filled circles) mice. Partial re-directed differentiation occurred in both MHC-I and -II selected cells. *P < 0.05, **P < 0.01, and ***P < 0.001(unpaired *t*-test)

To further address activation status of *Thpok* locus in *Bcl11b* mutant cells, we examined chromatin accessibility in DP thymocytes prepared from a *Bcl11b*ᴴᴹ/ᶠˡ:*Lck-Cre* and *Bcl11b*ᴴᴹ/ᴴᴹ:*Lck-Cre* mice by ATAC-seq[37]. Interestingly, the *Thpok* locus in *Bcl11b*ᴴᴹ/ᴴᴹ pre-selection thymocytes retained open chromatin signatures around the P1-promoter, resembling those seen in B cells expressing P1-derived transcripts (Fig. 3e). ChIP-seq analyses for Bcl11b binding in thymocytes of *Thpok*ᵍᶠᵖ:ᐃᵀᴱˢᴾᴱ/ᵍᶠᵖ:ᐃᵀᴱˢᴾᴱ mice detected association of Bcl11b at intronic regions nearby the P1-promoter (Fig. 1d), which retained accessibility in the *Bcl11b*ᴴᴹ/ᴴᴹ cells and whose sequences are evolutionarily conserved (Fig. 1d). Similar open structure was observed in DP cells of *Bcl11b*ᴴᴹ/ᶠˡ:*Lck-Cre* mice. These observations indicate that truncation of Bcl11b C-terminal sequences interferes, most specifically, with normal patterns of *Thpok* expression at early thymocyte differentiation, reconfiguring its regulome to mimic expression patterns found in B cells. However, RNA-seq did not detect significant elevation of B cell signature genes in *Bcl11b*ᵐ/ᵐ pre-selection thymocytes (Supplementary Fig. 3a, b), suggesting that Bcl11b might be involved in modulating chromatin accessibility at the restricted loci such as *Thpok*.

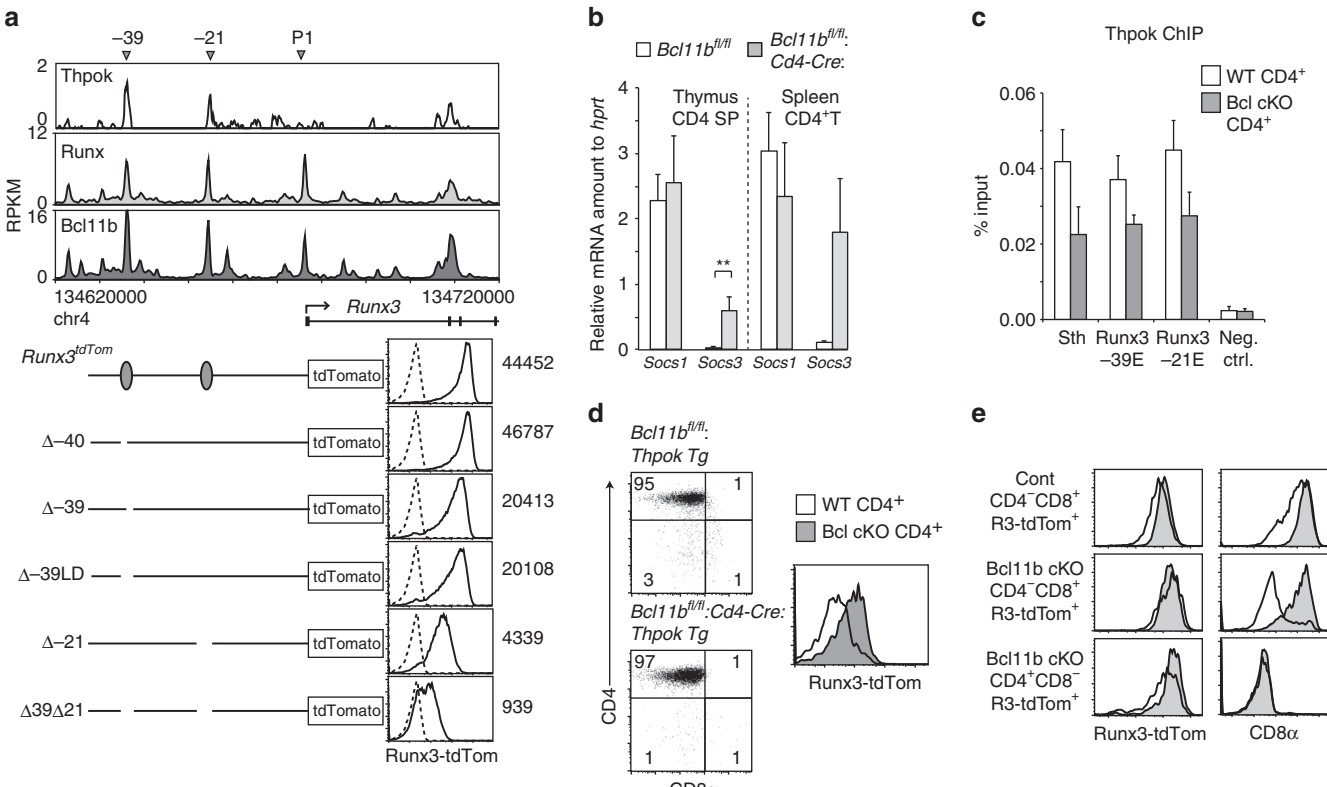

**Fig. 5** Roles of Bcl11b in ThPOK-mediated repression of *Runx3*. **a** Runx and Bcl11b ChIP-seq tracks at the *Runx3* locus using total thymocytes, along with a ThPOK ChIP-chip track using CD4 SP thymocytes. Schematic structure of mutant *Runx3^{tdTomato}* reporter alleles harboring several types of deletion (*bottom panel, left*). Histograms (*bottom panel, right*) showing Runx3-tdTomato expression in CD4⁻CD8⁺ thymocytes from indicated mutant alleles (*solid line*) and non-transgenic allele (*dotted line*). Data shown are representative of at least three independent mice. **b** Amount of *Socs1* and *Socs3* mRNA in CD4 SP thymocytes and CD4⁺ T cells from mice of indicated genotypes. Summary of three independent experiments. Mean ± SD **P < 0.01 (unpaired *t*-test). **c** ChIP-qPCR analyzing ThPOK binding to the *Sth* in *Thpok* and the −39E and −21E regions in the *Runx3* using chromatin from CD4⁺ T cells of Wt (white bars) and *Bcl11b^{f/f}:Cd4-Cre* (gray bars) mice. Combined data of three independent ChIP experiments. Means ± SD. **d** Flow cytometry analyzing the effect of a *Thpok* transgene on *Cd8* and *Runx3* expression. *Dot plots* showing CD4/CD8 expression in splenic T cells. Histograms at the right show *Runx3^{tdTomato}* reporter expression in Wt (white) and *Bcl11b-deficient* (gray) CD4⁺CD8⁻ subset. One representative of three individual mice. **e** Histograms showing CD8 and Runx3-tdTomato expression in control CD4⁻CD8⁺, Bcl11b-deficient CD4⁻CD8⁺ and CD4⁺CD8⁻ T cells after retroviral transduction of Wt (*open*) and the non-functional *HD* mutant ThPOK (*shaded*). One representative of two independent experiments

**Bcl11b is needed for lineage fidelity by MHC-restriction.** Contrary to CD4-skewing from *Bcl11b^{m/m}* progenitor, a previous study reported emergence of CD8⁺ T cells following *Bcl11b* inactivation at the DP stage using a *Cd4-Cre* driver. We reasoned that the apparent discrepancy could result from a stage-specific requirement for Bcl11b in regulating *Thpok* gene. We therefore next examined effect of *Bcl11b* inactivation by *Cd4-Cre* driver on *Thpok* expression. Although Thpok-GFP was uniformly de-repressed in all *Bcl11b^{m/m}* pre-selection thymocytes (Fig. 2c), expression of this reporter was variegated and detected in only half of the pre-selection thymocytes from *Bcl11b^{f/f}:Cd4-Cre* mice (Fig. 4a). We also observed that CD4⁺CD8⁺ and CD4⁻CD8⁺ subsets were present in mature thymocytes from *Bcl11b^{f/f}:Cd4-Cre* mice (Fig. 4a, b). It is noteworthy that, after positive selection, levels of Thpok-GFP in CD4⁺CD8⁻ cells differentiated from *Bcl11b^{m/m}* progenitors or in *Bcl11b^{f/f}:Cd4-Cre* mice were lower than those in control counterparts (Fig. 2c and Fig. 4a middle), a shift that was also reflected in attenuated *P2-Thpok* mRNA (Fig. 3c). Thus, loss of Bcl11b function disrupts not only *Thpok* repression, but also distinct mechanisms for its activation. These results suggests that, in a portion of pre-selection thymocytes in the *Bcl11b^{f/f}:Cd4-Cre* mice, the kinetics of *Thpok* de-repression are delayed compared to *Bcl11b^{m/m}* progenitors, which is combined later with low *Thpok* expression levels, allowing the cells to differentiate into CD4⁺CD8⁺ and CD4⁻CD8⁺ subsets.

Since previous studies showed release of *Runx3* repression in CD4⁺ T cells that differentiated under low levels of ThPOK[27, 38], we next examined *Runx3* expression using a *Runx3-tdTomato* reporter allele that monitors distal P1-promoter activity, which is CD8-linege specific and serves as a major source for Runx3 protein expression in T cells[27]. Importantly, we found that most mature thymocytes from *Bcl11b^{f/f}:Cd4-Cre* mice co-expressed Runx3-tdTomato and Thpok-GFP, regardless of their CD4/CD8 expression profiles (Fig. 4a), whereas expression of these genes was mutually exclusive between CD4⁺CD8⁻ and CD4⁻CD8⁺ cells from control mice. Contrary to a previous report showing *Runx3* de-repression in CD4⁺CD8⁺ DP thymocytes of *Bcl11b^{f/f}:Cd4-Cre* mice[31], Runx3-tdTomato was not detectably expressed in CD24^{hi}TCRβ^{lo} pre-selection thymocytes (Fig. 4a). Co-expression of Thpok-GFP and Runx3-tdTomato was observed also in CD4⁺CD8⁻ and CD4⁻CD8⁺ cells in the peripheral T cell pool of *Bcl11b^{f/f}:Cd4-Cre* mice (Fig. 4c), as well as in CD4⁺ T cells differentiated from *Bcl11b^{m/m}* progenitors (Fig. 4d). Thus, lineage-specific expression of two major factors, ThPOK and Runx3, which drive the CD4/CD8 lineage dichotomy, requires Bcl11b, or more specifically, Bcl11b function mediated by its C-terminal sequences.

We next examined the MHC specificity of T cells differentiated in *Bcl11b^{f/f}:Cd4-Cre* mice by crossing with MHC-I or MHC-II deficient mice. In mature thymocytes and peripheral T cell

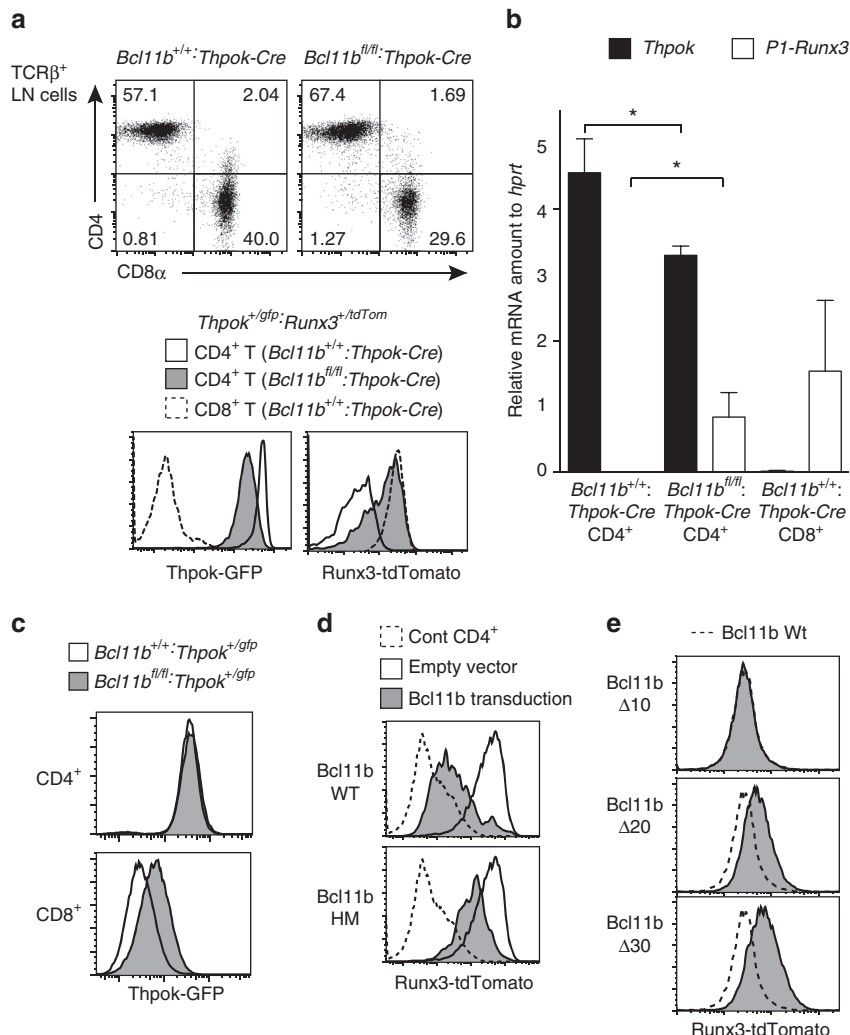

**Fig. 6** Bcl11b function requires its C-terminal zinc-finger motif. **a** Flow cytometry analyzing the expression of CD4, CD8, Thpok-GFP, and Runx3-tdTomato in peripheral T cells from *Bcl11b^{+/+}:Thpok-Cre* and *Bcl11b^{fl/fl}:Thpok-Cre* mice harboring *Thpok^{gfp}* and *Runx3^{tdTomato}* reporter genes. Inactivation of *Bcl11b* during the maturation of CD4-lineage cells by a *Thpok-Cre* transgene resulted in impaired Thpok-GFP expression, as well as Runx3-tdTomato de-repression. **b** RT-qPCR analyses showing relative amounts of *Thpok* (black bars) and *P1-Runx3* (white bars) mRNA in splenic CD4⁺CD8⁻ T cells of *Bcl11b^{+/+}:Thpok-Cre* and *Bcl11b^{fl/fl}:Thpok-Cre* mice. Results in splenic CD4⁻CD8⁺ T cells of *Bcl11b^{+/+}:Thpok-Cre* are shown as references. Data were summary of three experiments. Means ± SD *P < 0.05 (unpaired *t*-test). **c** Histograms showing Thpok-GFP expression from CD4⁺ and CD8⁺ T cells of *Bcl11b^{+/+}:Thpok^{+/gfp}* (white bars) and *Bcl11b^{fl/fl}:Thpok^{+/gfp}* (gray bars) mice 4 days after retroviral Cre transduction. One representative of three experiments. **d** Histograms showing Runx3-tdTomato expression 3 days after retroviral transduction of Wt and hypomorphic (HM) Bcl11b into peripheral CD4⁺CD8⁻Runx3-tdTomato⁺ cells from *Bcl11b^{fl/fl}:Runx3^{+/tdTomato}:Thpok-Cre* mice. Expression of Runx3-tdTomato in activated control CD4⁺ T cells (*dotted line*) after transduction with an empty vector (*open*) are shown as controls. **e** Histograms showing Runx3-tdTomato expression after retroviral transduction of mutant Bcl11b proteins lacking the C-terminal zinc-finger motif are shown as in **d**. The *dotted line* represents Runx3-tdTomato expression after transduction of Wt Bcl11b. One representative of at least two experiments **d**, **e**

populations, a significant proportion of both MHC-I- and MHC-II-specific cells in *Bcl11b^{fl/fl}:Cd4-Cre* mice expressed surface markers for the alternative lineage, CD8 and CD4, respectively (Fig. 4e, f). We therefore conclude that a substantial number of both MHC-I and MHC-II selected thymocytes failed in their specification for the appropriate developmental pathway, presumably due to spurious expression of ThPOK and Runx3 following positive selection. Ultimately, the chaotic expression of these factors resulted in "lineage scrambling", unlinking DP precursor thymocyte commitment from MHC specificity.

**Functions of Bcl11b in *Runx3* regulation**. To understand how Bcl11b regulates *Runx3* expression, we examined whether genomic regions around the *Runx3* locus were occupied by Bcl11b, Runx,

and ThPOK factors. In addition to the P1-promoter, Bcl11b and Runx bound at two regions (−39 and −21 kb), which were also bound by ThPOK (Fig. 5a) that is essential for *Runx3* repression in CD4⁺ T cells[22, 27]. To examine potential of these regions as *cis*-regulatory elements for *Runx3* regulation, we removed the regions separately or in combination from the *Runx3-tdTomato* allele using CRISPR/Cas9 technology[39] (Supplementary Fig. 4). Levels of Runx3-tdTomato expression in CD4⁻CD8⁺ SP thymocytes were reduced upon removal of either the −39 or −21 kb region, but not a control region of similar length positioned at −40 kb region (Fig. 5a). Expression of the *Runx3* reporter was further attenuated following combined deletion of the −39 and −21 kb regions. These observations indicated essential regulatory functions for these two regions in driving proper levels of *Runx3* expression in CD8⁺ T cells. The residual expression when both regions were

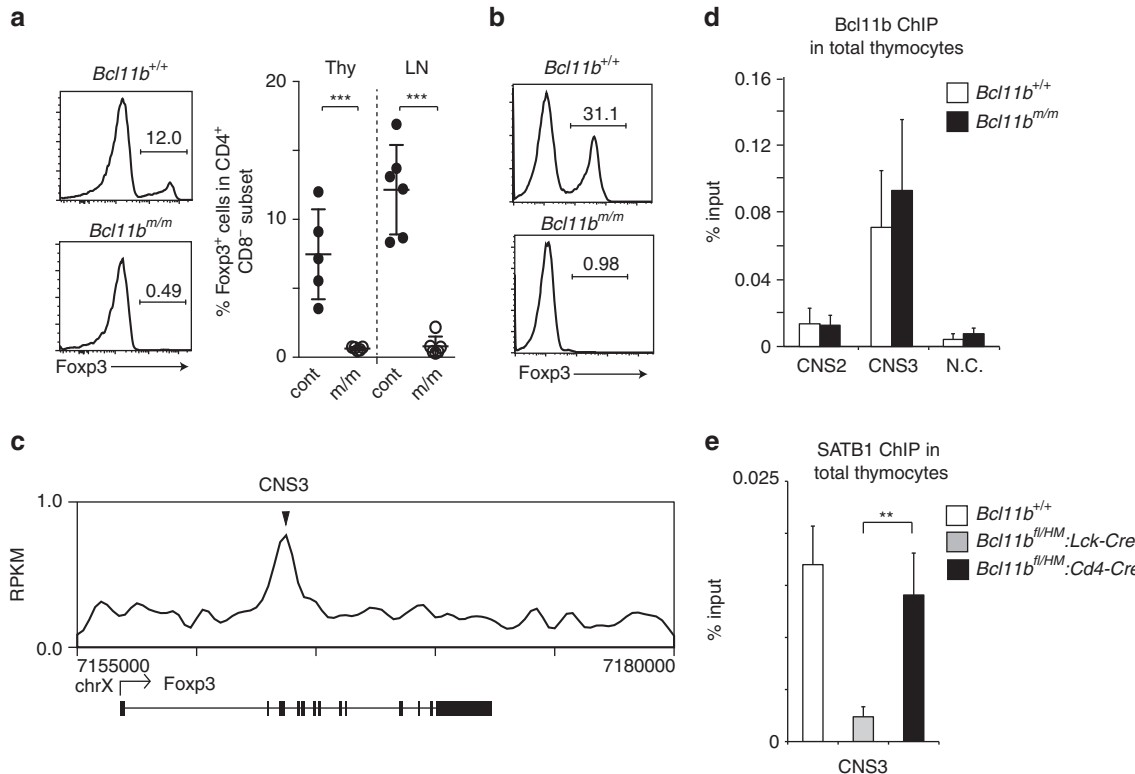

**Fig. 7** Regulation of *Foxp3* expression by Bcl11b. **a** Expression of Foxp3 in CD4+CD8− mature thymocytes differentiated in Rag1-KO hosts from *Bcl11b*+/+ and *Bcl11b*m/m fetal liver progenitors. A representative histogram (*left*) along with a summary of five independent hosts (*right*) are shown. ***P < 0.001 (unpaired *t*-test). **b** Histogram showing Foxp3 induction from CD62L+CD4+ T cells after in vitro culture under Treg differentiation condition. One representative of two experiments. **c** Bcl11b ChIP-seq tracks at the *Foxp3* gene in DP thymocytes. Gene structure, transcriptional orientation and position of conserved non-coding sequence (CNS) 3 are indicated. **d** Summary of ChIP-qPCR assay for binding of Wt (white bars) and hypomorphic (black bars) Bcl11b to CNS2 and CNS3 regions in the *Foxp3* locus in neonatal total thymocytes. **e** ChIP-qPCR for binding of SATB1 to the CNS3 enhancer in the *Foxp3* gene. Chromatin was collected from DP thymocyte precursors using mice with the indicated genotypes. Combined data from three independent ChIP experiments are shown. **P < 0.01 (unpaired *t*-test)

removed also indicated the presence of other enhancer(s) in the *Runx3* locus.

SOCS family proteins were proposed to mediate ThPOK-dependent repression of *Runx3* following thymocyte positive selection[29]. However, expression of *Socs1* and *Socs3* was not reduced in Bcl11b-deficient CD4+ T cells expressing *Runx3* (Fig. 5b). In Bcl11b-deficient cells, ThPOK bindings to −39 and −21 enhancers were also observed, albeit to a lesser extent, presumably due to lower ThPOK expression levels (Fig. 5c). We then tested whether ectopic ThPOK expression restored *Runx3* repression in CD4+ T cells. Although transgenic ThPOK prevented the development of CD8+ T cells in *Bcl11b*fl/fl:*Cd4-Cre* mice, Runx3-tdTomato was not repressed as much in CD4+CD8− splenic T cells (Fig. 5d). Consistent with this observation, retroviral ThPOK transduction into peripheral CD8+ T cells reduced Runx3-tdTomato expression in control cells, but not in Bcl11b-deficient cells, while CD8 expression was reduced in both cells, as reported previously[40] (Fig. 5e). These results indicated that full *Runx3* repression by ThPOK required Bcl11b after ThPOK bound to Runx3 enhancers. In contrast, ThPOK could repress *Cd8* expression in the absence of Bcl11b.

**Bcl11b regulates *Thpok* and *Runx3* expression**. To address at which developmental stages Bcl11b is necessary for appropriate *Thpok* and *Runx3* expression, we inactivated the *Bcl11b* gene during maturation of MHC-II selected cells using a *Thpok-Cre*

driver[41]. While CD4 and CD8 expression on peripheral T cells remained unchanged in *Bcl11b*fl/fl:*Thpok-Cre* mice, the expression of Thpok-GFP and Runx3-tdTomato in CD4+ T cells was decreased or increased, respectively (Fig. 6a, b). When *Bcl11b* was inactivated in differentiated CD8+ T cells by retroviral Cre transduction, de-repression of Thpok-GFP was also observed (Fig. 6c). These findings indicated that Bcl11b was necessary to maintain lineage-specific expression of both *Thpok* and *Runx3* after primary commitment of post-selection thymocytes to CD4/CD8 lineages.

We also tested the converse scenario, namely, whether retroviral Bcl11b transduction restored Runx3-tdTomato repression in differentiated CD4+ cells of *Bcl11b*fl/fl:*Thpok-Cre* mice. Levels of Runx3-tdTomato was reduced upon Bcl11b transduction compared with empty vector controls (Fig. 6d); however, transduction with a vector encoding the Bcl11bHM protein repressed Runx3-tdTomato to lesser extent than wild-type Bcl11b (Fig. 6d). To gain more insights into the domainal structure of Bcl11b, we tested the activities of other mutants generated by sequential deletion on its C-terminus (Supplementary Fig. 5). While a Bcl11b mutant retaining the final zinc-finger domain (Bcl11b-Δ10) and wild-type Bcl11b equally repressed Runx3-tdTomato expression, other mutants lacking the zinc-finger structure showed weaker activity (Fig. 6e). These results not only confirmed that loss of the last zinc-finger motif, rather than acquisition of aberrant sequences, attenuated Bcl11b function in the Bcl11bHM protein, but the new data also indicated that the

last zinc-finger motif is essential for Bcl11b function in preserving lineage-specific Runx3 expression.

In this regard, we compared structures of Bcl11 family proteins in other species. There are two zinc-finger clusters in Bcl11b, double zinc-fingers in its central region and triple zinc-fingers at its C-terminus. Comparative genomics analyses revealed that some species, such as the Ascidians (C. intestinalis) and Worm (C. elegans), lacked the middle zinc-finger motif in Bcl11 orthologues, whereas the triplet zinc-finger structure at the C-terminal was highly conserved during evolution (Supplementary Fig. 6), suggesting that C-terminal triplet zinc-fingers could be involved in ancestral and conserved function of Bcl11 family proteins.

**Stage-specific requirement for Bcl11b in Foxp3 transcription.** Additional analyses of T cell development from $Bcl11b^{m/m}$ progenitors detected a striking defect in the development of Foxp3[+] regulatory T cells (Treg) (Fig. 7a). In line with this finding, we were unable to induce Foxp3 expression in the mutant CD4[+] T cells even under potent Treg differentiation conditions in vitro (Fig. 7b). Contrary to this finding, a previous study reported the emergence of Foxp3[+] Treg cells following Bcl11b inactivation at the DP stage using a Cd4-Cre driver[8]. This apparent discrepancy could result from a direct and stage-specific requirement for Bcl11b in activating Foxp3, as we observed in Thpok regulation by Bcl11b. Recent work reported that the CNS3 region in the Foxp3 gene is essential to initiate Foxp3 activation by poising the Foxp3 promoter in an active state prior to the DP stage[42, 43]. Our ChIP-seq detected Bcl11b binding to CNS3 in total thymocytes (Fig. 7c) and our ChIP-qPCR detected similar binding of both Bcl11b and Bcl11b[HM] proteins to CNS3 (Fig. 7d). Since we recently observed that binding of the genome organizer, SATB1, to CNS3 is essential for Foxp3[+] Treg development in the thymus[41, 44], we examined SATB1 association with CNS3 in Bcl11b mutant thymocytes. Consistent with development of Foxp3[+] cells, SATB1 binding to CNS3 was unaffected in total thymocytes from $Bcl11b^{fl/HM}$:Cd4-Cre mice. On the contrary, SATB1 recruitment was significantly decreased in those cells from $Bcl11b^{fl/HM}$:Lck-Cre mice (Fig. 7e). These results suggested that priming of the Foxp3 locus by Bcl11b prior to the DP stage is essential for activation of Foxp3 gene via promoting SATB1 recruitment to the CNS3 enhancer.

## Discussion

In this study, we demonstrate that a hypomorphic Bcl11b protein (Bcl11b[HM]), lacking only its final zinc-finger domain, supports T-lineage commitment and ILC2 development. In contrast, the hypomorphic Bcl11b mutation fails to control the expression of transcription factors, including ThPOK, Runx3, and Foxp3, which are important for specification of mature T lineages (CD4-helper, CD8-cytotoxic, and Treg, respectively). Importantly, our findings dissect the functional architecture of Bcl11b, revealing independent regions that are responsible for early T-lineage commitment versus transcriptional regulation of lineage-specifying genes during positive selection.

The distinct effects on expression of Thpok and Foxp3 genes of the germline hypomorphic mutation when compared with a DP stage-specific inactivation of its gene indicate that Bcl11b acts on these two genes at an early developmental stage, presumably at the DN to DP transition, to control appropriate expression of these lineage-specifying transcription factors at later stage. For Thpok, while a half of precursors still repressed Thpok gene after conditional loss of Bcl11b expression, hypomorphic Bcl11b mutation caused Thpok expression in all precursors that retains non-T cells like open chromatin structure around the

P1-promoter. This observation indicates that early Thpok repression at the DN to DP transition by Bcl11b involves chromatin closing. Our results using the $Thpok^{gfp:\Delta TESPE}$ mutant reporter allele lacking Sth silencer revealed that Bcl11b regulates such chromatin closing independently of the Sth, which is essential to silence Thpok expression at later stage specifically in cytotoxic T cells[35]. Given that the Sth is sufficient to repress reporter transgenes driven by heterologous enhancer/promoter in DP thymocytes[20] and is essential to prevent expression of the endogenous gene in Bcl11b-sufficient DP cells[19], both Sth-independent and Sth-dependent pathways operate to repress Thpok in precursor DP thymocytes. Considering that ectopic ThPOK expression in DP precursor thymocytes not only disturbs helper/cytotoxic lineage choice[16] but also increases a risk for lymphomagenesis[45], it might be beneficial to have two independent mechanisms to secure no leaky ThPOK expression in DP precursors. It is noteworthy that Sth-independent Thpok repression was reversible, since Thpok-GFP expression from the $Thpok^{gfp:\Delta TESPE}$ allele was induced in mature T cells, suggesting that irreversible epigenetic changes such as DNA methylation are unlikely to be involved in early Sth-independent Thpok repression by Bcl11b. Given that Sth could stably silence Thpok transcript, particularly from the P2-promoter, in CD8-lineage cells[35], mechanisms of Sth-dependent and Sth-independent Thpok repression could be different.

The precise involvement of Bcl11b in Sth-independent Thpok repression remains elusive. As detected by ChIP-seq, Bcl11b covers regions around the P1-promoter in an Sth-independent manner. Along with evolutional conservation in sequences of intronic region just downstream of the P1-promoter, there might exist uncharacterized regulatory element(s) that suppresses P1-promoter activity upon expression of Bcl11b. Binding of mutant Bcl11b[HM] protein to such intronic region suggest that the C-terminal zinc-finger motif in Bcl11b is necessary to close P1-promoter region after Bcl11b binding to putative regulatory element(s). Alternatively, Bcl11b may be involved indirectly in the Sth-independent Thpok repression through an uncharacterized intermediate factor, however, the similar gene expression profiles between wild-type and $Bcl11b^{m/m}$ precursors disfavors this possibility.

De-repression of Thpok in mature CD8[+] populations that emerge from $Bcl11b^{fl/fl}$:Cd4-Cre precursors suggests that Bcl11b is required also for Sth-mediated Thpok repression. Thus, Bcl11b regulates Thpok repression by both Sth-independent and Sth-dependent mechanisms. Based on our data, we presume that Bcl11b acts initially to inhibit P1-promoter activity in an Sth-independent manner. Either sequentially or simultaneously, Bcl11b primes three T cell specific regulatory regions, Sth and two enhancers, TE and PE. Upon such priming, Sth initiates to suppress TE and PE activity, thereby removal of Sth results in Thpok de-repression in Bcl11b-sufficent precursors in which TE/PE enhancers also becomes functional after Bcl11b-mediated priming. In this regard, an Sth-independent mechanism alone is likely to be insufficient for extinguishing TE/PE enhancer activities. Thus, we assume that pre-conditioning of T cell specific regulatory regions by Bcl11b is an essential process for dissecting TCR signals that arise from engagement by distinct MHCs, and for converting them into establishment of "on" and "off" Thpok expression status for inducing helper- or cytotoxic-fate via inactivation or maintaining Sth activity, respectively.

Bcl11b similarly primes the Foxp3 locus for later activation upon exposure to agonistic selection signals[46]. A recent study showed that the pioneering enhancer, CNS3[43], confers a poised state to the Foxp3 promoter in precursor cells in response to TCR stimuli[42]. In the present study, we show that binding of SATB1 to CNS3 in precursors, an essential process for activation of Foxp3

gene in the thymus[41, 44], requires Bcl11b-mediated priming before transition to the DP stage. Whether common Bcl11b-mediated mechanisms operate to prime the *Thpok* and *Foxp3* loci remains unclear; however, we establish that the C-terminal zinc-finger motif of Bcl11b is involved in both processes. Collectively, our results reveal a set of novel functions for the T-lineage commitment factor Bcl11b, which "pre-conditions" precursors for integration of environmental cues, specifically TCR signals, into a developmental program that dissects effector lineages and shapes the primary T cell pool.

Another striking finding from our studies is that the lack of Bcl11b inhibits both the positive and negative regulatory regions near *Thpok*. One possible model to explain how Bcl11b controls multiple regulatory regions would be its function as a scaffold to facilitate communication between the distal elements and the *Thpok* promoters. Using insertion ChIP technology[47], we recently observed that the *Sth* and *PE* regions were positioned in close proximity during differentiation into CD8-lineage cells (I.T. manuscript in preparation). Thus, it is possible that Bcl11b is involved in shaping promoter-regulatory elements communications on the *Thpok* locus.

Contrary to *Thpok* regulation, Bcl11b is merely involved in repression of *Runx3* in CD4[+] T cells[22, 27]. Prior studies invoked an indirect mechanism for *Runx3* repression by ThPOK via the induction of SOCS family proteins[29]. However, our finding that ThPOK associates with functional *Runx3* enhancers raises the possibility that ThPOK directly antagonizes enhancer function. Indeed, ThPOK bound to the enhancers even in Bcl11b-deficient cells that were de-repressing *Runx3* albeit normal level of *Socs* family gene expression. These data suggest that the antagonistic action of ThPOK against *Runx3* expression requires Bcl11b after its binding to the *Runx3* enhancers. It is conceivable that Bcl11b assists ThPOK in preventing the formation of looping between the enhancers and the P1-Runx3 promoter. Nevertheless, it will be important in future studies to examine whether lineage-specific chromatin structures are formed in *Thpok* and *Runx3* loci and whether Bcl11b regulates these topologies.

The $C_2H_2$ zinc-finger motif not only serves as a DNA binding domain that recognizes specific sequences, but also as a docking module for RNA and proteins[48]. We found that the Bcl11b[HM] protein retains its ability to associate with regulatory regions, indicating that its final zinc finger is dispensable for DNA binding. However, it remains possible that the other four zinc-fingers domains have compensatory functions in this regard. There are two zinc-fingers clusters in mammalian Bcl11 family proteins, double zinc-fingers in its central portion and a trio of zinc-fingers at its C-terminus. Although there are Bcl11-related proteins in lower species that lacks the central dual zinc-finger domain, a trio of zinc-fingers at the C-terminus is evolutionally well conserved. Our results indicate that this domain is essential for gene regulation, presumably through modulating topological structures. Bcl11a has been shown to play a key role in the switch from fetal γ- to adult β-globin in human[49]. It is conceivable that common Bcl11 family-dependent mechanisms play a key role in developmentally programmed switching of regulatory regions by modulating topological structures through the C-terminal triplet zinc-finger stretch. Isolation of interacting molecules with it will further elucidate regulatory mechanism by Bcl11 family.

## Methods
**Mice**. Runx1[Δ446] mice[50], Runx3[fl] mice[51], Bcl11b[+/−] mice[36], Thpok[gfp] mice[22], Thpok[gfp:241-401RM] mice[35] and Thpok transgenic mice[19] have been described. Lck-Cre mice, and Cd4-Cre mice were from Dr J. Takeda, and Dr C. Wilson, respectively. β2m-deficient mice (stock No:002070), Rag1-deficient mice (Stock No:002216) were from Jackson laboratory, and I-Aβ deficient mice and Il2rg[−/−]: Rag1[−/−] were from Taconic. In order to generate a Thpok[gfp:ΔTESPE] allele, TE/Sth and PE regions were sequentially removed from the Thpok[gfp] reporter allele by using homologous recombination in ES cells. In order to construct the target vector for Bcl11b[fl] allele, genomic fragment isolated from the phage library (Stratagene) was used as starting material. The neomycin-resistance gene (neo[r]) flanked with two loxP sequences was cut out from pL2-Neo(2) vector and thymidine kinase (TK) gene was isolated from pNT vector. A 5′ short homology region was amplified by PCR and was ligated with the neo[r] fragment, followed by sequential ligation to add 3′ homology region. The third loxP sequences placed in the 3′ untranslated region (UTR) were derived from synthetic oligo-nucleotide. The single nucleotide deletion that causes a frameshift mutation in the Bcl11b[m] allele was accidentally incorporated into the target vector at some point during above sequential ligation steps. To construct a real target vector for Bcl11b[fl] mutation, correct exon 4 sequences were replaced in the Bcl11b[m] target vector. A fragment that harbors the identical Bcl11b[m] single nucleotide deletion but lacks the third loxP sequences was used to construct the target vector for Bcl11b[HM] allele. In order to generate Runx3[tdTomato] reporter allele, we constructed the targeting vector by replacing YFP cDNA fragment with cDNA fragment encoding tdTomato fluorescent protein in pRx31-KIN cassette vector[27], which was used to generate the Runx3[YFP] allele[27]. Those targeting vectors were transfected into M1 ES cells as previously described[22]. ES clones underwent homologous recombination were identified by PCR with appropriate primers sets. In order to delete −39 kb and −21 kb genomic regions from the Runx3[tdTomato] reporter allele by CRISPR/Cas9 technology[52, 53], we selected two single guide RNA (sgRNA) target sequences that flanks target genomic regions, which were shown in Supplementary Fig. 3. Custom sgRNA, in which CRISPR RNAs was fused to a normally trans-encoded tracrRNA, were transcribed from a T7 promoter in the pUC18 vector by in vitro transcription with MEGAshortscript T7 kit (Life Technologies, AM1354) and both the Cas9 mRNA and the sgRNAs were purified using MEGAclear kit (Life Technologies, AM1908). The sgRNAs and mRNA encoding Cas9 were co-injected into cytoplasm of fertilized eggs that were prepared by in vitro fertilization with Runx3[tdTomato/tdTomato] sperm and wild-type oocytes. Cas9-mediated double-stranded DNA breaks resolved by non-homologous end joining (NHEJ) ablated the intervening sequences. Founder off-spring that had deletion of the target genome region were crossed to wild-type mice, and F1 founders that harbor both deletion mutation and Runx3[tdTomato] allele were selected for establishing mouse line and were analyzed. In order to generate allele harboring double deletions at −39 kb and −21 kb regions, sgRNA pair used for generation of Runx3[Δ21E] mutation were injected into eggs obtained by in vitro fertilization between Runx3[tdTomato:Δ39LD/+] sperm and wild-type oocytes. RNA injection was performed by the Animal Facility Group at RIKEN, IMS. All mice were maintained in the specific pathogen free animal facility at the RIKEN IMS, and all animal procedures were in accordance with institutional guidelines for animal care and with the protocol (28-017) approved by the safety section in RIKEN Yokohama Campus.

**DNA pull-down assay**. In order to analyze binding of Bcl11b to *Thpok* silencer (*Sth*) core sequences in vitro, nuclear extract was prepared from $10 \times 10^7$ total thymocytes and were mixed with 10 μg of biotin-labeled synthetic oligo-nucleotides probe (Eurofins Genomics) in 400 μl of affinity purification (AP) buffer (20 mM HEPES-KOH (pH 7.5), 80 mM KCl, 10% Glycerol, 0.1% Triton X, 0.5 mM PMSF, 1× cOmplete protease inhibitor Cocktail tablets (Roche)) on ice for 3 h, followed by mixture with 100 μl of Dynabeads M-280 Streptavidin (Thermo Fisher Scientific) for additional 3 h on ice. After one time wash with AP buffer, beads were incubated with 500 pmol of mutant non-labeled oligo-nucleotide for 30 min at 4 °C, and were washed with AP buffer three times. Captured protein complexes by beads were then released into 50 μl of SDS sample buffer and were applied for western blot analyses. Sequences for oligo-nucleotides probes used were as follows:

Wt:5′-TGGCAGCCACCGCCTCTTCAGGTGGGTTGGGCGGTCG CGGTAGGGGTTCTGGGGGGCGGCGGGAGGGAGGGGGCTGCGGT CTGAG-3′

Mutant:5′-TGGCAGCCACCGCCTCTTCAGCACCCATGGGCGCAG GCGGTAGGGGTTCTGGGGGGCGGCGGGAGGGAGGGGGCACGCC ACTGAG-3′.

**Co-immune precipitation and western blot**. Nuclear extract was prepared from $10 \times 10^6$ total thymocytes with NE-PER kit (Thermo Fisher Scientific) in the presence of cOmplete protease inhibitor cocktail tablets (Roche), followed by incubation with 2 μg of anti-Bc11b (A300-383A, Bethyl Laboratories) antibody, which was pre-mixed with 50 μl of Dynabeads ProteinA, overnight at 4 °C with gentle rotation. Antibody bound proteins were released into 20 μl of SDS sample buffer and was loaded onto SDS-PAGE gel. In western blot, anti-Runx1 serum[54] and antibodies that reacts with N-terminal sequences (A300-383A, Bethyl Laboratories) and C-terminal sequences (A300-385A, Bethyl Laboratories) of Bcl11b were used to detect protein.

**Chromatin immunoprecipitation**. For ChIP-seq, $3 \times 10^7$ thymocytes that were freshly prepared from C57BL/6 mice at 4–6 weeks old were washed once with PBS supplemented with 1% FCS, cOmplete protease inhibitor cocktail (Roche) and 1

mM 4-(2-Aminoethyl) benzenesulfonyl fluoride hydrochloride (Sigma-Aldrich), and were cross-linked by incubation in a 1% paraformaldehyde solution for 10 min with gentle rotation at RT. The reaction was stopped by addition of glycine to 0.15 M. Cells were then washed with ice-cold PBS containing 1% FCS for 10 min with gentle rotation at 4 °C, and were lysed in Lysis Buffer 1 (50 mM HEPES pH 7.5, 140 mM NaCl, 1 mM EDTA, 10% Glycerol, 0.5% NP-40, 0.25% Triton X-100) supplemented with cOmplete protease inhibitor cocktail tablets (Roche) for 10 min at 4 °C with gentle rotation. Nuclei were pelleted and were washed by Lysis Buffer 2 (10 mM Tris-HCl pH 8.0, 200 mM NaCl, 1 mM EDTA and 0.5 mM EGTA) supplemented with cOmplete protease inhibitor (Roche). Pelleted chromatin was resuspended in 300 μl of Lysis Buffer 3 (10 mM Tris-HCl pH 8.0, 100 mM NaCl, 1 mM EDTA, 0.5 mM EGTA, 0.1% Sodium deoxycholate and 0.5% N-laurylsarcosine sodium salt), and was sonicated using a model XL2000 ultrasonic cell disruptor (MICROSON) at output level 6 for 15 s for 10 times. After removing debris by centrifugation, 30 μl of 10% Triton X-100 (Nacalai tesque) was added to 270 μl of supernatant (final 1%) and sonicated chromatin was incubated overnight at 4 °C with 5 μg of anti-Bcl11b rabbit polyclonal antibody (A300-383A, Bethyl Laboratories) that was pre-conjugated with 50 μl of Dynabeads M-280 Sheep anti-Rabbit IgG (Thermo Fisher Scientific). After washing beads with ChIP-RIPA (50 mM HEPES (pH 7.6), 500 mM LiCl, 1 mM EDTA, 1% NP-40, 0.7% sodium deoxycholate) and TE buffer supplemented with 50 mM NaCl, immunoprecipitates were eluted from beads into 100 μl of elution buffer (50 mM Tris-HCl pH 8.0, 10 mM EDTA, 1% SDS) by incubation for 15 min at 65 °C with vigorous shaking. Eluted immunoprecipitates were then incubated at 65 °C overnight for reverse-crosslinking. Input DNA and ChIP DNA were treated with RNAse A (Thermo Fisher Scientific) at 37 °C for 1 h, followed by incubation with Proteinase K (Thermo Fisher Scientific) at 55 °C in the presence of 6 mM CaCl$_2$ for one hour. DNA was purified by Phenol/Chloroform extraction for ChIP-seq or ChIP DNA Clean and Concentrator kit (ZYMO RESEARCH) for ChIP-qPCR. For ChIP-seq analysis, purified DNA was subjected to re-sonication with a Covaris S220 to produce DNA fragments with an average size of 200 bp, and was used for library construction with NEBNext ChIP-seq Library Prep Master Mix set for Illumina Kit (NEB). Sequencing was performed by the RIKEN IMS sequence facility with Illumina HiSeq 1500 or at Hiroshima University using a GAIIx (illumina). To detect ThPOK binding regions in the murine *Runx3* locus, we performed ChIP-on-chip experiment. Chromatin DNA isolated from $10 \times 10^6$ CD4$^+$ SP thymocytes from FH-ThPOK mice[22] was immunoprecipitated with anti-Flag (M2; Sigma-Aldrich), and were hybridized against custom microarrays generated by Agilent that tiled through the murine *Runx3* locus up to ~ 565 kb upstream and ~ 66 kb downstream by means of 60-nucleotide probes. Probe hybridization and scanning of oligo-nucleotide array data were done according to manufacturer's protocol (Agilent). For analytical ChIP, we used 5 μg of rabbit anti-Bcl11b antibody (A300-383A, Bethyl Laboratories), 5 μg of rabbit anti-ThPOK polyclonal antibody[27] and 5 μg of rabbit anti-SATB1 polyclonal antibody (ab70004, Abcam). Quantitative PCR was performed using the StepOnePlus Real-Time PCR system (Applied Biosystems) with SYBR Green detection system. Primers sequences for quantitative PCR are listed in the Supplementary Table 1.

**Flow cytometry analyses**. Thymus, spleen, and lymph nodes were removed from mice at 4–8 weeks of age, and were mashed through a 70 μm cell strainer to make single-cell suspensions. Cells were stained with following antibodies purchased from BD-Bioscience: CD4 (RM4-5), CD8 (53-6.7), CD24 (M1/69), CD25 (PC61), CD154 (MR1), IL1RL1 (DJ8), ScaI (E13-161.7), KLRG1 (2F1), and TCRβ (H57-597). Foxp3 staining buffer Set (00-5523-00) and anti-mouse FoxP3 antibody (FJK-16s) from affymetirx eBioscience was used to stain Foxp3. Antibodies were used at a concentration of 2.5 μg ml$^{-1}$. Multi-color flow cytometry analysis was performed using a FACSCANTO II (BD-Bioscience) and data were analyzed using FlowJo (Tree Star) software. Cell subsets were sorted using a FACSAria II (BD Biosciences) by cell sorting facility at RIKEN IMS.

**In vitro T cell culture**. Purified T cells were cultured in custom ordered Dulbecco's Modified Eagle Medium (D-MEM, KOHJIN BIO) supplemented with 10% heat inactivated FBS (Hyclone). $1.0 \times 10^6$ cells were stimulated in 24-well plates pre-coated with 2 μg ml$^{-1}$ anti-CD3e antibody (553058, BD Bioscience) with 2 μg ml$^{-1}$ soluble anti-CD28 antibody (553295, BD Bioscience) for two days. For induction of Treg differentiation, 10 ng ml$^{-1}$ TGFβ1 (7666-MB, R&D systems), 5 μg ml$^{-1}$ anti-IL4 (554433, BD Biosciences) and 5 μg ml$^{-1}$ anti-IFNγ (554409, BD Biosciences) were added throughout the culture.

**Reconstitution of T cell development in host mice**. Total liver cells with either *Bcl11b*$^{+/+}$ or *Bcl11b*$^{m/m}$ genotype were prepared from 13.5 to 14.5 dpc embryos, and were suspended in 400 μl of D-MEM supplemented with 10% heat inactivated FBS (Hyclone). 200 μl of these fetal liver cells were intravenously injected into sub-lethally irradiated (6.5 Gy, Gammacell 40 Exactor, MDS Nordion) Rag1-deficeint or I-Aβ deficient mice that expressing Ly9.1. For ILC2 reconstitution, *Il2rg*$^{-/-}$:*Rag1*$^{-/-}$ mice were used as recipients. Host mice were supplied with antibiotics (1 mg ml$^{-1}$ neomycin and 100 unit ml$^{-1}$ polymyxin B) containing water for first 2 weeks, and were analyzed 8 weeks after the fetal liver cell injection. In I-

Aβ deficient host mice, donor derived lymphocytes were identified as Ly9.1-negative cells.

**RNA isolation, RT-qPCR and RNA-seq**. After total cellular RNA extraction by using Trizol reagent (Thermo Fisher Scientific), samples were incubated with RNase-free DNase I (Thermo Fisher Scientific) before cDNA synthesis with the SuperScriptIII First Strand Synthesis System (Invitrogen). Quantitative RT-PCR was performed using the StepOnePlus Real-Time PCR system (Applied Biosystems) with an internal fluorescent TaqMan probe, Universal ProbeLibrary (Roche) or SYBR Green detection system. Primers sequences and probe sequences for quantitative RT-PCR are listed in the Supplementary Table 2. Primer sequence to measure total *Thpok* mRNA was previously described[16]. For RNA-seq analyses, purified $1.0 \times 10^6$ cells were used to prepare mRNA and 200–300 ng total RNA were used for the library construction with SureSelect Strand Specific RNA-Seq LibraryPreparation kit (Agilent Technologies) according to the manufacturer's protocol. Sequencing was performed by genomics facility at RIKEN IMS with Illumina HiSeq 1500.

**Assay of transposase-accessible chromatin sequencing**. Assay of transposase-accessible chromatin sequencing (ATAC-seq) samples were prepared from 25,000 sorted cells. The transposase reactions were carried as previously described[55] with 12 or 13 total PCR cycles. Amplified DNA fragments were purified with QIAGEN MinElute PCR Purification Kit and size-selected twice with Agencourt AMPure XP. Libraries were quantified with KAPA Library Quantification Kit for Illumina Sequencing Platforms (KAPA Biosystems), and size distribution was checked on a Bioanalyzer (Agilent High Sensitivity DNA chip, Agilent Technologies). Libraries were sequenced on Illumina HiSeq 2500.

**Retroviral transduction into T cells and T cell culture**. Retroviral vectors encoding Cre recombinase or ThPOK in pMSCV-GFP vector was previously described[56]. Retroviral vectors encoding mutant Bcl11b proteins were generated by insertion of corresponding cDNA into pMSCV-GFP vector after conformation of sequences of PCR amplified fragment. Those vectors were transfected into Plat-E packaging line (a gift from Dr T. Kitamura at the University of Tokyo) by FuGENE6 Reagent (Promega, E2691) and supernatant collected two days after transfection were used for transduction into activated T cells by in vitro TCR stimulation with immobilized anti-CD3 and soluble anti-CD28 by spin infection at 2400 r.p.m. for 60 min in the presence of 4 μg of polybrene (Sigma-Aldrich). Transdcuced T cells were identified according to GFP expression for FACS analyses and RNA extraction.

**Data and statistical analyses**. Sequences retrieved in ChIP-seq experiments and ATAC-seq were aligned on the mouse genome (mm9) using the bowtie2 (http://bowtie-bio.sourceforge.net/index.shtml) with default parameters, and accumulated reads were normalized using total mapped reads. RNA-seq reads were mapped using tophat2 (https://ccb.jhu.edu/software/tophat/index.shtml) to the mouse genome and gene expression was estimated with fragments per mapped reads per kilobase (FPKM) values calculated using cufflinks (http://cole-trapnell-lab.github.io/cufflinks/) and gene annotation provided by iGenome. Primary component analysis (PCA) was carried out using our in-house program with FPKM values of gene sets whose annotation was provided by Gene Ontology database (http://geneontology.org/). Statistical analysis was performed by F-test and unpaired *t*-test with or without Welch's correction using using GraphPAd Prism6 (Graphpad software). Figures display means and SD.

**Comparative genomics analyses**. Homologous sequences of Bcl11 family were searched with human Bcl11b protein sequences as a query against sequences listed at the Ensemble genome browser 85 and Genebank with "TBLASTN" algorithm. As for the Japanese lamprey search, Bcl11 protein sequences for human, mouse, and elephant shark were used as queries to search the Japanese lamprey genome assembly (http://jlampreygenome.imcb.a-star.edu.sg/) with "TBLASTN" algorithm. Genomic regions that showed high similarity were extracted and searched against the non-redundant protein database at NCBI to confirm their identity. Alignments of C-terminal sequences of Bcl11b orthologue of each species were performed by a computer program Clustal Omega (http://www.clustal.org/omega/) with default options for amino acid sequences using zinc-finger regions obtained proposed by BLASTP algorithm and NCBI conserved domain database (https://www.ncbi.nlm.nih.gov/Structure/cdd/wrpsb.cgi).

**Data availability**. RNA-seq, ChIP-seq and ATAC-seq data that support the findings of this study have been deposited with accession GSE90134, GSE90949, and GSE90989, respectively. All other relevant data are available from the authors.

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

## Acknowledgements

We thank C. Miyamoto and R. Chihara for mouse genotyping, N. Yoza for cell sorting, T. Ishikura for ES cell aggregation, Y. Iizuka for microinjection of RNA for CRISPR/Cas9 mediated genome editing. We are grateful to Dr Thomas Boehm for helpful suggestion on analyses of lamprey genome and Dr Eugene Oltz for critical reading of the manuscript. This work was supported by JSPS KAKENHI Grant Number JP 21229008 (I.T.), JP 26293109 (I.T.) and JP 22021045 (I.T.), the Uehara Memorial Foundation (I.T.), Takeda Science Foundation (I.T.) the National Institutes of Health grant R01AI 097244-01A1 (T.E.).

## Author contributions

S.K., H.T., W.S., K.K., Y.N., and I.T. performed experiments for phenotypic characterization of *Bcl11b* mutant mice. M.T. and K.M. performed reconstitution experiments. S.M. isolated ES cell clones. T.A.E. performed statistical and comparative genomics analyses. K.N. made mutant Bcl11b expression vectors. S.K., A.K., and T.I. performed

sequencing of ChIP libraries. T.E. provided anti-ThPOK antibody. Y.L. and A.M. per-
formed ATAC-seq. B.V. analyzed Japanese lamprey genome. Y.K. and R.K. provided
*Bcl11b* germline KO mice. I.T. designed experiments, interpreted data and wrote the
manuscript.

## Additional information

**Competing interests:** The authors declare no competing financial interests.

**Reprints and permission** information is available online at http://npg.nature.com/
reprintsandpermissions/

**Publisher's note:** Springer Nature remains neutral with regard to jurisdictional claims in
published maps and institutional affiliations.

