## [Peer Review File · Nature Communications]

Reviewers' comments:

Reviewer #1 (Thymocyte, ThPOK/RUNX3)(Remarks to the Author):

The transcription factors ThPOK and Runx3 control the differentiation of MHC class-II restricted CD4 and MHC class-I restricted CD8 T cells, respectively. How TCR signaling couples with the lineage-specific transcriptional program is critical for the understanding of CD4/CD8 lineage decision and their different immune function. In this manuscript, Satoshi Kojo et al found that the T-lineage commitment factor Bcl11b is essential for the expression of both ThPOK and Runx3. Bcl11b ablation resulted in the random expression of these factors and disordered CD4/CD8 differentiation. In addition, the authors demonstrated that the ThPOK repression by Bcl11b in pre-selection thymocytes is independent of silencer Sth. The observation of this manuscript is interesting and the experiments are clearly described in general.

Minor revision:

1. This study showed that the differentiation of CD4/CD8 T cells was disconnected from TCR restriction by MHC in the absence of Bcl11b, however, how Bcl11b translates TCR signaling to control the expression of ThPOK and Runx3 (and thereafter lineage-specific transcriptional program) remains largely unclear. It would be very helpful to analysis the Bcl11b expression in DN1-4, DP and CD4, CD8 cells as well as MHC class-I and MHC class-II TCR signaled cells.

2. Bcl11b is able to bind multiple regions on ThPOK gene and play a role on the chromatin accessibility; the authors should compare the levels of DNA methylation, histone acetylation between WT and Bcl11b deficient cells or at least explain more how this works in the Discussion part.

3. The regulation of Bcl11b on Foxp3 expression feels like a distraction, and this part could be organized as supplementary information.

Reviewer #2 (Thymocyte, hematopoietic lineage)(Remarks to the Author):

This manuscript presents a complex set of discoveries about the role for Bcl11b during the segregation of CD4 and CD8 T-cell lineages during positive selection in the thymus. These two fates are governed, respectively, by the ThPOK and Runx3 transcription factors that are activated in a selective way depending on whether the developing thymocytes have been positively selected by interactions with class II or with class I MHC. While Bcl11b is needed to generate the DP cells that are eligible to undergo this fate choice, a hypomorphic Bcl11b allele allows cells to reach positive selection but then to undergo an abnormal version of positive selection in which ThPOK and Runx3 are activated inappropriately. It had previously been reported that conditional Bcl11b deletion at the DP stage led to precocious activation of ThPOK and Runx3, but now Kojo et al. analyze the phenotype in much greater depth. They find that full-length Bcl11b is required to restrict ThPOK expression to class II MHC-selected cells and to restrict Runx3 expression to class I selected cells, and they show that the Bcl11b effect on ThPOK is exerted not only through recognized enhancer and silencer elements but also by suppressing activation through additional, previously undiscovered cis-regulatory elements. They show that Bcl11b knockout cells activate ThPOK in fact through elements that are normally used primarily by non-T cells. Finally, they dissect the Bcl11b structural requirements for these activities and show that they depend on the integrity of the C-terminal Zinc Finger of the protein, which they show to be part of a highly conserved Zinc Finger cluster with likely orthologues even in *C. elegans*.

The work is extremely extensive, thoughtful, and impressive. It is framed by an excellent introduction and discussion. However, the phenotypes are complicated to appreciate. Some readers may become

confused enough not to be sure whether they are convinced by the logic of the work. One problem is that ThPOK and Runx3 affect the main stage markers for thymocyte development, CD4 and CD8, so that for the scrambled phenotype cells with inappropriate expression of ThPOK and Runx3 it is not clear what the true normal counterparts may be. The other problem is that ThPOK and Runx3 negatively regulate each other under normal conditions, and so it is complicated to explain how the abnormal expression of one may or may not be related to the abnormality in the other when Bcl11b is missing. Finally, because of the number of groups that have contributed to this work, it is not surprising that there are some rough parts in the organization of the paper, for example, experiments done with mutants that have not been described yet at that point in the paper. Point by point suggestions for the authors follow.

1. The whole paper needs to be supported with statistics for the different results, both in the legends and in the text. Right now the results given are mostly examples from data from one of two experiments. A more complete analysis needs to be provided and statistical significance of the effects seen needs to be reported.

2. Structural issue in the paper: the Foxp3 story appears to be out of context in this paper and interrupts the flow. It actually creates confusion because it raises questions of whether the cells that would normally be Treg precursors are ever formed normally in the first place, which is in doubt considering that so many of the supposed CD4 cells generated in the Bcl11b knockouts are not legitimate CD4 lineage. Also, the Bcl11b binding to the CNS3 element is much weaker than to the elements shown in Figs 1 & 3, raising questions about its significance. Finally, while the use of Lck-Cre vs. CD4-Cre is very helpful to distinguish early and later acting roles of Bcl11b on Foxp3, this makes the reader realize that the ThPOK and Runx3 stories are not analyzed this way. It thus creates a sense that something useful might be missing from the main subject of the paper. It seems that this Foxp3 story would be better separated out from the rest of the paper. Without it, there is a smooth logical connection from Fig. 3 to Fig. 5.

3. On p. 6, the complex regulatory elements of the ThPOK gene are introduced very briefly as though we already know about them from previous work. Many readers will not, and the diagram in Fig. 1d leaves out the positions of TE and P2. Furthermore, the Sth knockout reporter allele is not diagrammed or fully described in text, methods, or figure legend. It would be helpful to include more background here on all three of these points.

4. In characterizing the mutant phenotypes in Supplementary Fig.1, Fig. 2, and Fig. 5, it is very important to provide actual cell counts for the total thymocyte populations and the important subsets, not just flow cytometry profiles. There are very likely cell viability and proliferation effects here that have a substantial effect on how much of the phenotype should be interpreted as due to excessive maturation, to trans-differentiation, or to selective cell death. The cell death and population size effects from the Bcl11b deletion, in particular, need to be compared directly with any effects from the hypomorph.

5. In Fig. 2, the complicated effects on CD4/CD8 phenotype should be explained here, where they first appear, to help the reader to get through the remainder of the paper. How much of the CD4/CD8 phenotype should the reader take seriously as an indication of what the cells "should" have been?

6. Fig. 3 shows very interesting effects on differential promoter use but again is hard to interpret. This section of the text on pp.8-9 is confusingly written.

a. One baseline fact that seems uncertain is how much of total ThPOK in normal DP thymocytes comes from P1 and how much from P2.

b. Where the text states that P2 expression is dependent on the PE, it should clarify immediately whether the regulatory elements for P1-driven expression are known or not.

c. Looking at Fig. 3a, it seems that normal CD4 T cells express ThPOK equally from both promoters, so it is not clear why the text says that there is "aberrant P1-promoter usage in Bcl11b(m/m) thymocytes". Instead, it appears that there is an aberrant lack of P2 promoter usage in these cells. But if this is not true, it should be explained better.

d. The Bcl11b binding to a (del)TESPE allele is shown in Fig. 1d without discussion at that point,

whereas in Fig. 3 an ATAC-seq profile is shown for the Bcl11b(m/m) cells that appears to be similar. Are these open peaks the same ones that Bcl11b would bind in the absence of TESPE?

e. The Bcl11b mutant that is used in Fig. 3 is suddenly the “HM” mutant, which is only defined in relation to Fig. 4. This should be explained in relation to Fig. 3.

7. On p. 11, in discussing the phenotype of Bcl11b conditional knockout mutants, the authors suggest that delayed kinetics of ThPOK de-repression are related to the effects seen. But everything shown in Figs. 1-3 seems to suggest that ThPOK is activated more in Bcl11b mutants than in wildtype, rather than delayed. Is there any evidence that ThPOK activation is actually delayed instead of accelerated? It seems that another explanation could be found, anyway, based on the authors’ previous work showing Bcl11b as a component of the Runx repression complex on CD4. Is it not possible that ThPOK and Runx3 lose their ability to cross-repress each other in the absence of functional Bcl11b, through defective protein complex formation? The coexpression of Runx3-tdTomato and Thpok-GFP in such a large proportion of cells is one of the most exciting results in the whole paper.

8. The authors’ results seem to indicate some inferences that would be good to state explicitly, one way or the other. For example, does the Bcl11b(HM/HM) genotype support CD4 T cell development better than the Bcl11b mutant? This seems likely because others have reported severe defects in CD4 T cell populations in Bcl11b conditional knockout mice. Is it true that Runx3 cannot repress ThPOK unless wildtype Bcl11b is present?

9. Minor points:

a. Fig. 3 must be labeled better to show genotypes in panels a, b, and c. If the whole point is that panel C shows Bcl11b(m/m) genotypes only, this is a critical piece of information to interpret the figure. Fig. 3c would be much more informative if it showed normal DP cells for comparison.

b. If Fig. 4 is kept in the paper, even if it is moved to the Supplement, it must be better labeled. The y axes in Fig. 4c is not labeled as ATAC-seq. Cell numbers are not provided. The CD4/CD8 phenotype of the cells analyzed in Fig. 4d is not reported. Also, since Fig. 4d is the only figure panel in the whole paper that is about Satb1, it is vital to label it as Satb1 ChIP – this is not Bcl11b ChIP like all the other ChIP data shown.

c. On p. 7, clarify that the Thpok(gfp) allele is gene-disruptive, to help explain why heterozygotes are used. But could there also be a haploinsufficiency effect?

d. On p. 11, line 6, indicate “(Fig. 2c and Fig. 5a, RIGHT)”. Fig. 5a actually has two panels’ worth of information in it.

e. On p. 11, second line from the bottom, write “was not DETECTABLY expressed in CD24(hi)...”. The RNA analysis of ref. 30 might have been sensitive to lower-level early activation of Runx3 than the protein fluorescent reporter here.

f. Fig. 6 is unfortunately labeled in such a tiny font that this important evidence is hard to see.

g. In Fig. 7, when Bcl11b transduction downregulates Runx3, how high is the Bcl11b expression as compared to wildtype cells with normal Bcl11b?

Reviewer #3 (Thymocyte, hematopoietic lineage)(Remarks to the Author):

The manuscript by Kojo, et al describes a thorough genetic analysis that indicates an important role for the Bcl11b transcription factor in CD4/CD8 lineage commitment during T cell development in the thymus. Previous studies had documented an essential function for Bcl11b during T lineage commitment of early thymic precursors, but to date, a thorough study of Bcl11b at later stages of thymic development had not been performed. This study utilizes a series of Bcl11b alleles along with several different cre-drivers to demonstrate an important role for Bcl11b in the proper regulation of ThPOK and Runx3, two key factors in CD4/CD8 lineage commitment. Specifically, the data show both early and late requirements for Bcl11b in this process, as well as a dissection of key Bcl11b domains, and important regulatory elements in the ThPOK and Runx3 loci.

This is an extremely interesting manuscript that describes a detailed and comprehensive examination of Bcl11b functions in CD4/CD8 lineage commitment during thymocyte development. The data are important and, once suitably revised, would make an excellent contribution to the field. Currently, there are several concerns the authors should address:

1. On a general note, the authors should consider the caveat that expression of the Bcl11b^{m/m} allele throughout T cell development might have altered pre-TCR β -selection stage thymic progenitors. The fact that these mice still have DP thymocytes does not necessarily demonstrate that all earlier stages of T cell development are normal. Therefore, there is always the concern that effects seen at later stages of T cell development are a reflection of earlier alterations. This caveat should be discussed in the manuscript.

2. More importantly, the data shown throughout the manuscript lack all indications of absolute cell numbers. This information is critical to interpreting the results presented. For many of the experiments, the authors conclude that developing mature thymocytes are experiencing 'lineage-scrambling', implying that a cell which is receiving external signals to differentiate into one lineage is 'confused' and instead, is mis-expressing genes representative of both lineages. For this conclusion to be valid, the numbers of cells in the different genotypes of mice should be similar. Alternatively, if some genotypes of mice have very few thymocytes altogether, the flow cytometry data showing relative proportions of cells with various phenotypes would not really support the conclusions the authors are arriving at.

3. In many figures, certain panels of thymocytes are labeled 'mature'. The figure legends do not clearly indicate what this means. Are all panels of 'mature' thymocytes gated on TCR β ⁺? What about Fig. 2E, which is not labeled 'mature' – are these gated on TCR β ⁺? If all analysis was done using TCR β ⁺ as the only criterion, what about other lineages of TCR β ⁺ thymocytes beyond conventional CD4 and CD8 T cells, such as NKT cells or other MHC class Ib-specific cells? This is likely only a major concern if total thymocyte numbers are vastly different between mice of different genotypes, but should be addressed at some level.

4. In general, the figure legends contain insufficient information. Given word limitations on manuscripts, this is somewhat understandable, but it is often difficult to decipher exactly what the data in a given figure are showing.

5. The manuscript is not clear on which experiments are done with unmanipulated mice (not fetal liver chimeras) and which were done by reconstitution, and furthermore, whether any were done in neonatal versus adult mice. It seemed the authors stated that the Bcl11b^{m/m} mice died shortly after birth, yet these mice were used in many of the figures (Fig 2, 3, 4, 5). What were these mice - neonates? This information needs to be more clearly presented.

6. Some of the findings are less convincing than others. For example, the data from Bcl11b^{fl/fl} x CD4-cre MHC I-null mice shown in Figure 5D are not at all straightforward. Many of the cells are DN, rather than CD8⁺, as argued. Also, the reduced expression of Runx3-tdTom in CD8SP thymocytes transduced with ThPOK (Figure 6e) is also quite modest. Is this difference biologically significant?

REVIEWERS' COMMENTS:

Reviewer #1 (Remarks to the Author):

The authors have addressed all my concerns in the revised manuscript. No further questions.

Reviewer #2 (Remarks to the Author):

The authors have made extremely useful responses to the reviewers' comments and have strengthened the clarity and power of this manuscript. It is an impressive and important contribution that reveals a great deal about how Bcl11b participates in different cell fate decisions, and along the way it sheds unexpected light on what the underlying logic of those decisions must be. For example, the difference in timing between the roles of Bcl11b in controlling Thpok and in controlling Runx3 expression is very interesting, as is the stark difference between the roles of Bcl11b C-terminal domains in ILC2 development, in DN cell commitment, and in regulation of Thpok and Runx3.

Small typos and minor issues that authors (or production editors) may wish to address:

--The new Fig. 4f should be mentioned in the text. One possible place is in line 248.

--On p. 13, regarding Fig. 5d, the reduction of Runx3-tdTomato by ThPOK seems weak even in control cells, limiting the dynamic range over which Bcl11b effects can be measured. Perhaps on line 275 it would be helpful to say "but not AS MUCH in Bcl11b-deficient cells", rather than the current wording which implies a more dramatic difference.

--In Fig. 3d, correct spelling of label to "DN3 thymocytes". Also, in Fig. 3e, shouldn't the y axes be RPM rather than RPKM for ATAC-seq?

--In Fig. 5d, it would be helpful to add genotype labels to the CD4/CD8a flow cytometry panels.

Reviewer #3 (Remarks to the Author):

The authors have satisfactorily addressed the concerns raised in the original reviews. The revised manuscript is substantially improved, and is recommended for publication.

** See Nature Research's author and referees' website at www.nature.com/authors for information about policies, services and author benefits

Reviewers' comments:

Reviewer #1

The transcription factors ThPOK and Runx3 control the differentiation of MHC class-II restricted CD4 and MHC class-I restricted CD8 T cells, respectively. How TCR signaling couples with the lineage-specific transcriptional program is critical for the understanding of CD4/CD8 lineage decision and their different immune function. In this manuscript, Satoshi Kojo et al found that the T-lineage commitment factor Bcl11b is essential for the expression of both ThPOK and Runx3. Bcl11b ablation resulted in the random expression of these factors and disordered CD4/CD8 differentiation. In addition, the authors demonstrated that the ThPOK repression by Bcl11b in pre-selection thymocytes is independent of silencer Sth. The observation of this manuscript is interesting and the experiments are clearly described in general.

We appreciate the positive comments.

Minor revision:

1. This study showed that the differentiation of CD4/CD8 T cells was disconnected from TCR restriction by MHC in the absence of Bcl11b, however, how Bcl11b translates TCR signaling to control the expression of ThPOK and Runx3 (and thereafter lineage-specific transcriptional program) remains largely unclear. It would be very helpful to analysis the Bcl11b expression in DN1-4, DP and CD4, CD8 cells as well as MHC class-I and MHC class-II TCR signaled cells.

We thank the reviewer #1 for this suggestion. Analyses of the *Bcl11b* gene expression pattern by using a knock-in reporter strain was already reported in cited reference #5 (Liu et al. Science 2010, 329, 85-89) and in the recent manuscript published in Nature Immunology (Kueh et al. at Nat Immunol. 2016, 17:956). We also generated our own Bcl11b reporter strain and observed the same expression pattern as in the above published results (Figure R1 for reviewers). Basically, Bcl11b expression is not dramatically changed during T cell development. Since published data are already available for *Bcl11b* expression, we refer to these two papers to describe the Bcl11b expression pattern (page3). In this study, we show that regulation through the C-terminus zinc-finger motif in Bcl11b protein is important to control T cell development. Thus, the Bcl11b expression is unlikely to serves as a central regulatory mechanism to translate TCR signals.

Figure R1. Expression of *Bcl11b* during T cell development. Expression of *Bcl11b* was tracked by reporter hCD2 expression from a *Bcl11b-IRES-hCD2* allele.

2. Bcl11b is able to bind multiple regions on ThPOK gene and play a role on the chromatin accessibility; the authors should compare the levels of DNA methylation, histone acetylation between WT and Bcl11b deficient cells or at least explain more how this works in the Discussion part.

We thank the reviewer #1 for this suggestion. It is always a profound question of how transcription factors regulate target gene expression, but this question is not always easily addressed. In this study, we show that *Thpok* repression by Bcl11b at the transition into DP stage is independent of the *Thpok* silencer (*Sth*). As pointed out by the reviewer, Bcl11b binds to the *Thpok* gene at the conserved intronic region. We agree that understanding of the molecular mechanisms that modulate chromatin accessibility in *Thpok* is important. However, little is known about the physiological relevance of DNA methylation and histone acetylation in *Thpok* regulation. We are now addressing the roles of such a Bcl11b-bound region by removing it from the mouse genome using genome editing technology. We think that the better approach to this question will be to analyze epigenetic modifications of this novel *Thpok* locus together with *Bcl11b* mutant cells. However, it will take another few months to get homozygous mutant mice. In order to discuss how *Sth*-independent *Thpok* repression by Bcl11b may be regulated, we added sentences on page16 to point out that *Sth*-independent *Thpok* repression functions in a reversible manner and therefore irreversible epigenetic changes are unlikely to be involved in this repression. We also discuss the possible involvement of the intronic region in *Sth*-independent and Bcl11b-dependent *Thpok* repression in the third paragraph of the discussion.

3. The regulation of Bcl11b on Foxp3 expression feels like a distraction, and this part could be organized as supplementary information.

We thank the reviewer for this suggestion. Reviewer #2 made a similar suggestion. Taking these suggestions into account, we reorganized the manuscript structure and show only essential results regarding *Foxp3* regulation by Bcl11b in Figure 7 of the revised manuscript. A stage-specific requirement for Bcl11b in *Thpok* regulation is a major finding in this work. Since we think it is important to be able to generalize this Bcl11b function, we believe that it is worth showing some data about another target gene that is regulated by Bcl11b similarly in a stage-specific manner.

In Figure 7, we show that the phenotype in FoxP3⁺ Treg cell generation was different in the two *Bcl11b* mutant mouse models, the germline hypomorphic mutation and the conditional *Bcl11b* inactivation by *Cd4-Cre*, as was observed in the phenotypes in CD4/CD8 lineage choice. We also show that early Bcl11b function at the transition into the DP stage is important for efficient recruitment of SATB1 to the pioneering enhancer CNS3 in the *Foxp3* gene in DP thymocytes. We believe that these FoxP3⁺ Treg phenotype results are not distracting, rather, they are informative and strengthen the manuscript.

Reviewer #2 (Thymocyte, hematopoietic lineage)(Remarks to the Author):

This manuscript presents a complex set of discoveries about the role for Bcl11b during the segregation of CD4 and CD8 T-cell lineages during positive selection in the thymus. These two fates are governed, respectively, by the ThPOK and Runx3 transcription factors that are activated in a selective way depending on whether the developing thymocytes have been positively selected by interactions with class II or with class I MHC. While Bcl11b is needed to generate the DP cells that are eligible to undergo this fate choice, a hypomorphic Bcl11b allele allows cells to reach positive selection but then to undergo an abnormal version of positive selection in which ThPOK and Runx3 are activated inappropriately. It had previously been reported that conditional Bcl11b deletion at the DP stage led to precocious activation of ThPOK and Runx3, but now Kojo et al. analyze the phenotype in much greater depth. They find that full-length Bcl11b is required to restrict ThPOK expression to class II MHC-selected cells and to restrict Runx3 expression to class I selected cells, and they show that the Bcl11b effect on ThPOK is exerted not only through recognized enhancer and silencer elements but also by suppressing activation through additional, previously undiscovered cis-regulatory elements. They show that Bcl11b knockout cells activate ThPOK in fact through elements that are normally used primarily by non-T cells. Finally, they dissect the Bcl11b structural requirements for these activities and show that they depend on the integrity of the C-terminal Zinc Finger of the protein, which they show to be part of a highly conserved Zinc Finger cluster with likely orthologues even in *C. elegans*.

The work is extremely extensive, thoughtful, and impressive. It is framed by an excellent introduction and discussion.

We appreciate this very positive evaluation.

However, the phenotypes are complicated to appreciate. Some readers may become confused enough not to be sure whether they are convinced by the logic of the work. One problem is that ThPOK and Runx3 affect the main stage markers for thymocyte development, CD4 and CD8, so that for the scrambled phenotype cells with inappropriate expression of ThPOK and Runx3 it is not clear what the true normal counterparts may be. The other problem is that ThPOK and Runx3 negatively regulate each other under normal conditions, and so it is complicated to explain how the abnormal expression of one may or may not be related to the abnormality in the other when Bcl11b is missing. Finally, because of the number of groups that have contributed to this work, it is not surprising that there are some rough parts in the organization of the paper, for example, experiments done with mutants that have not been described yet at that point in the paper.

We thank the reviewer for raising these issues. We agree that the phenotypes observed in Bcl11b mutants are complicated and that inappropriate ThPOK and Runx3 expression, which are mutually exclusive in normal CD4⁺ and CD8⁺ cells, respectively, makes the phenotype more complicated. Thus, true normal counterparts are the CD4⁺Thpok⁺ helper and CD8⁺Runx3⁺ cytotoxic subsets. However, by using our own reporter mouse strains, we detected mature T cells expressing both ThPOK and Runx3. This is a real and important finding, and reflects the beauty of a genetic approach to a difficult problem. These data clearly show that Bcl11b is essential, not only to regulate lineage specific expression of Thpok and Runx3 genes, but also to establish the essential antagonistic interplay between ThPOK and Runx3 during thymocyte differentiation. As we interpret the major points of reviewer #2, we should increase the clarity of the manuscript. We addressed most this Reviewer's suggestions/criticisms and believe that the revised manuscript is significantly improved.

Point by point suggestions for the authors follow.

1. The whole paper needs to be supported with statistics for the different results, both in the legends and in the text. Right now the results given are mostly examples from data from one of two experiments. A more complete analysis needs to be provided and statistical significance of the effects seen needs to be reported.

We thank the reviewer for this suggestion. We added statistical information in figure legends and in the text. We also added statistical significance in Figure S1e, Figure 4a and Figure 4e.

2. Structural issue in the paper: the Foxp3 story appears to be out of context in this paper and interrupts the flow. It actually creates confusion because it raises questions of whether the cells that would normally be

Treg precursors are ever formed normally in the first place, which is in doubt considering that so many of the supposed CD4 cells generated in the Bcl11b knockouts are not legitimate CD4 lineage. Also, the Bcl11b binding to the CNS3 element is much weaker than to the elements shown in Figs 1 & 3, raising questions about its significance. Finally, while the use of Lck-Cre vs. CD4-Cre is very helpful to distinguish early and later acting roles of Bcl11b on Foxp3, this makes the reader realize that the ThPOK and Runx3 stories are not analyzed this way. It thus creates a sense that something useful might be missing from the main subject of the paper. It seems that this Foxp3 story would be better separated out from the rest of the paper. Without it, there is a smooth logical connection from Fig. 3 to Fig. 5.

We thank the reviewer for this suggestion, which was also pointed out by reviewer #1. As suggested, for a smooth logical connection, we moved data about Foxp3 regulation by Bcl11b into Figure 7 in the revised manuscript. As pointed out by this reviewer, the use of two Cre Tg models, Lck-Cre and CD4-Cre, is a very powerful approach to address stage-specific functions of Bcl11b and thus we wished to use the same approach to understand ThPOK and Runx3 regulation by Bcl11b. However, we first noticed that *Bcl11b^{fl/fl};Lck-Cre* mice had a severe reduction of DP thymocytes and nearly lacked mature thymocytes. These phenotypes prevented us from examining ThPOK/Runx3 expression in post-selection thymocytes. Given the emergence of mature thymocyte from *Bcl11b^{m/m}* progenitors, we thought that a combination of *Bcl11b* inactivation by *Lck-Cre* together with the *Bcl11b^m* mutant allele should be informative. To pursue this aim, we had to remove the loxP site in the 3' UTR of the *Bcl11b^m* allele, and for that reason we generated the *Bcl11b^{HM}* allele. We then examined the effect of *Bcl11b* inactivation by Lck-Cre in the context of the *Bcl11b^{HM}* allele (*Bcl11b^{HM/fl};Lck-Cre* mice). Unfortunately, we found a partial developmental block at the DN to DP transition and only a very tiny population of mature thymocytes in *Bcl11b^{HM/fl};Lck-Cre* mice. Probably, haploinsufficiency of Bcl11b^{HM} protein is not sufficient to support differentiation of mature thymocytes to the same extent as that from *Bcl11b^{m/m}* progenitors. Of even more concern, our detailed analysis of these mice revealed that the remaining mature thymocytes population in the *Bcl11b^{HM/fl};Lck-Cre* mice is heavily contaminated with a leaky population that escaped Cre-mediated *Bcl11b* inactivation. We therefore concluded that we could not practically utilize the *Lck-Cre* driver to analyze ThPOK and Runx3 expression in cells after positive selection. Instead, we utilized these two Cre Tg systems to compare chromatin accessibility in the *Thpok* gene in DP thymocytes (Figure 3e). Related to point 6e by the reviewer, we agree that the section in which Bcl11b-HM allele was described in the original manuscript was unclear. In the revised manuscript, we described the *Bcl11b^{fl}* and *Bcl11b^{HM}* allele after initial characterization of the *Bcl11b^m* allele.

3. On p. 6, the complex regulatory elements of the ThPOK gene are introduced very briefly as though we

already know about them from previous work. Many readers will not, and the diagram in Fig. 1d leaves out the positions of TE and P2. Furthermore, the Sth knockout reporter allele is not diagrammed or fully described in text, methods, or figure legend. It would be helpful to include more background here on all three of these points.

Thank you for pointing out this problem. In response, we indicated the position of TE and P2 in Figure 1d and marked the regions deleted in the *Thpok*^{gfp:ΔTESPE} alleles with shaded squares. We also added more background about the known regulatory elements in the *Thpok* gene in the introduction and explained in more detail how we generated the *Thpok*^{gfp:ΔTESPE} allele (the structure of this mutant allele is now diagrammed in Figure 3d).

4. In characterizing the mutant phenotypes in Supplementary Fig. 1, Fig. 2, and Fig. 5, it is very important to provide actual cell counts for the total thymocyte populations and the important subsets, not just flow cytometry profiles. There are very likely cell viability and proliferation effects here that have a substantial effect on how much of the phenotype should be interpreted as due to excessive maturation, to trans-differentiation, or to selective cell death. The cell death and population size effects from the *Bcl11b* deletion, in particular, need to be compared directly with any effects from the hypomorph.

According to this suggestion, we show reduced thymocyte cellularity in newborn *Bcl11b*^{m/m} mice in Figure S1 and cells numbers of MHC-I and MHC-II selected cells in Figure 4 in the revised manuscript.

5. In Fig. 2, the complicated effects on CD4/CD8 phenotype should be explained here, where they first appear, to help the reader to get through the remainder of the paper. How much of the CD4/CD8 phenotype should the reader take seriously as an indication of what the cells “should” have been?

According to this suggestion, we modified the text to explain more about CD4/CD8 expression for general readers.

6. Fig. 3 shows very interesting effects on differential promoter use but again is hard to interpret. This section of the text on pp.8-9 is confusingly written.

a. One baseline fact that seems uncertain is how much of total ThPOK in normal DP thymocytes comes from P1 and how much from P2.

We are sorry for this confusion. In normal DP thymocytes, there is no ThPOK expression at all.

b. Where the text states that P2 expression is dependent on the PE, it should clarify immediately whether the regulatory elements for P1-driven expression are known or not.

We added a sentence pointing out that TE is likely to drive *P1-Thpok* transcription.

c. Looking at Fig. 3a, it seems that normal CD4 T cells express ThPOK equally from both promoters, so it is not clear why the text says that there is “aberrant P1-promoter usage in *Bcl11b*(m/m) thymocytes”. Instead, it appears that there is an aberrant lack of P2 promoter usage in these cells. But if this is not true, it should be explained better.

We thank the reviewer for this suggestion. We revised the sentences to ‘unusual P1-promoter activation without P2-promoter activation in *Bcl11b*^{m/m} thymocytes’ in the revised manuscript.

d. The *Bcl11b* binding to a (del)TESPE allele is shown in Fig. 1d without discussion at that point, whereas in Fig. 3 an ATAC-seq profile is shown for the *Bcl11b*(m/m) cells that appears to be similar. Are these open peaks the same ones that *Bcl11b* would bind in the absence of TESPE?

Yes, the open peaks detected in *Bcl11b*^{m/m} cells by ATAC-seq are the same as the regions occupied by *Bcl11b* in the *Thpok*^{gfp:ΔTESPE} allele. We modified the text to clarify this point.

e. The *Bcl11b* mutant that is used in Fig. 3 is suddenly the “HM” mutant, which is only defined in relation to Fig. 4. This should be explained in relation to Fig. 3.

We thank the reviewer for pointing out this issue, which is related to point 2. Although we have described the logic behind the generation of the *Bcl11b*^{HM} allele in our reply to point 2, we think that a better place to describe the *Bcl11b*^{HM} allele for the first time in the manuscript is after the section about the initial characterization of *Bcl11b*^m allele (page 8). We hope that the reviewer#2 will agree that this modification has improved the flow of the manuscript.

7. On p. 11, in discussing the phenotype of *Bcl11b* conditional knockout mutants, the authors suggest that delayed kinetics of ThPOK de-repression are related to the effects seen. But everything shown in Figs. 1-3 seems to suggest that ThPOK is activated more in *Bcl11b* mutants than in wildtype, rather than delayed. Is there any evidence that ThPOK activation is actually delayed instead of accelerated?

We thank the reviewer for this criticism and apologize for our confusing description. What we mean by ‘delayed’ is following. While *Thpok* de-repression was observed in almost all *Bcl11b*^{m/m} pre-selection thymocytes, it occurred only in a proportion of pre-selection thymocytes of *Bcl11b*^{fl/fl}:*Cd4-Cre* mice. In the rest of these cells, *Thpok* de-repression is ‘delayed’ compared to *Bcl11b*^{m/m} cells. This later de-repression of *Thpok*, which was combined with lower levels of ThPOK, is likely to allow CD8 expression during thymocyte differentiation in *Bcl11b*^{fl/fl}:*Cd4-Cre* mice, resulting in emergence of

CD4⁺CD8⁺ and CD4⁻CD8⁺ subsets instead of the CD4-skewing observed in *Bcl11b*^{m/m} mutants. We have revised the text to increase clarity on this point (page 11).

It seems that another explanation could be found, anyway, based on the authors' previous work showing Bcl11b as a component of the Runx repression complex on CD4. Is it not possible that ThPOK and Runx3 lose their ability to cross-repress each other in the absence of functional Bcl11b, through defective protein complex formation? The coexpression of Runx3-tdTomato and Thpok-GFP in such a large proportion of cells is one of the most exciting results in the whole paper.

Yes, co-expression of Runx3-tdTomato and Thpok-GFP is a major finding in this study and we provided some mechanistic insights into why Runx3 is not repressed in the presence of ThPOK.

8. The authors' results seem to indicate some inferences that would be good to state explicitly, one way or the other. For example, does the Bcl11b(HM/HM) genotype support CD4 T cell development better than the Bcl11b mutant? This seems likely because others have reported severe defects in CD4 T cell populations in Bcl11b conditional knockout mice. Is it true that Runx3 cannot repress ThPOK unless wildtype Bcl11b is present?

We thank the reviewer for this inquiry, which is related to point 7. Previous studies and our current study showed a decrease of CD4 T cell subset in conditional Bcl11b knock out mice by Cd4-Cre *Bcl11b*^{fl/fl}:Cd4-Cre mice. However, the germline *Bcl11b*^{m/m} mutant mice showed CD4-skewing. As we explained in reply to the point 7, this difference would stem from different kinetics of Thpok de-repression between the two models. Given the complete developmental arrest at the DN2a stage due to the Bcl11b null-mutation, our hypomorphic Bcl11b mutation provided a unique opportunity to examine pre-selection thymocyte de-repressing Thpok. Our results clearly showed that *Thpok* is not repressed in Runx3-expressing cells and, vice versa, Runx3 is not repressed by ThPOK, without Bcl11b function. Please note that Thpok expression is induced even in CD8⁺ T cells upon removal of Bcl11b by retroviral Cre transduction (Figure 6C).

9. Minor points:

a. Fig. 3 must be labeled better to show genotypes in panels a, b, and c. If the whole point is that panel C shows Bcl11b(m/m) genotypes only, this is a critical piece of information to interpret the figure. Fig. 3c would be much more informative if it showed normal DP cells for comparison.

We added genotype information.

b. If Fig. 4 is kept in the paper, even if it is moved to the Supplement, it must be better labeled. The y axes

in Fig. 4c is not labeled as ATAC-seq. Cell numbers are not provided. The CD4/CD8 phenotype of the cells analyzed in Fig. 4d is not reported. Also, since Fig. 4d is the only figure panel in the whole paper that is about Satb1, it is vital to label it as Satb1 ChIP – this is not Bcl11b ChIP like all the other ChIP data shown.

We show only essential data regarding Foxp3 activation in Figure 7 in the revised manuscript with modified labels.

c. On p. 7, clarify that the *Thpok(gfp)* allele is gene-disruptive, to help explain why heterozygotes are used. But could there also be a haploinsufficiency effect?

Thpok^{gfp} is a gene-disruptive, knock-in reporter allele. A possible haploinsufficient effect is not excluded. However, since we used *Bcl11b^{+/+}:Thpok^{+gfp}* cells as a control and use the *Thpok^{gfp}* allele mainly to monitor *Thpok* expression, there should be no major problems. Indeed, we have data using *Bcl11b^{m/m}:Thpok^{+/+}* cells and confirmed that these cells show a similar CD4-skewing to that observed by *Bcl11b^{m/m}:Thpok^{+gfp}* cells.

d. On p. 11, line 6, indicate “(Fig. 2c and Fig. 5a, RIGHT)”. Fig. 5a actually has two panels’ worth of information in it.

e. On p. 11, second line from the bottom, write “was not DETECTABLY expressed in CD24(hi)...”. The RNA analysis of ref. 30 might have been sensitive to lower-level early activation of Runx3 than the protein fluorescent reporter here.

f. Fig. 6 is unfortunately labeled in such a tiny font that this important evidence is hard to see.

We have corrected these points accordingly.

g. In Fig. 7, when Bcl11b transduction downregulates Runx3, how high is the Bcl11b expression as compared to wildtype cells with normal Bcl11b?

We did not measure retroviral-vector derived Bcl11b levels in comparison to the Bcl11b level in normal CD4 T cells. Since the point of this retroviral expression is to compare functionality among several Bcl11b mutants, this information is not absolutely necessary.

Reviewer #3 (Thymocyte, hematopoietic lineage)(Remarks to the Author):

The manuscript by Kojo, et al describes a thorough genetic analysis that indicates an important role for the Bcl11b transcription factor in CD4/CD8 lineage commitment during T cell development in the thymus. Previous studies had documented an essential function for Bcl11b during T lineage commitment of early

thymic precursors, but to date, a thorough study of Bcl11b at later stages of thymic development had not been performed. This study utilizes a series of Bcl11b alleles along with several different Cre-drivers to demonstrate an important role for Bcl11b in the proper regulation of ThPOK and Runx3, two key factors in CD4/CD8 lineage commitment. Specifically, the data show both early and late requirements for Bcl11b in this process, as well as a dissection of key Bcl11b domains, and important regulatory elements in the ThPOK and Runx3 loci.

This is an extremely interesting manuscript that describes a detailed and comprehensive examination of Bcl11b functions in CD4/CD8 lineage commitment during thymocyte development. The data are important and, once suitably revised, would make an excellent contribution to the field.

We appreciate these positive comments.

Currently, there are several concerns the authors should address:

1. On a general note, the authors should consider the caveat that expression of the Bcl11bm/m allele throughout T cell development might have altered pre-TCRb-selection stage thymic progenitors. The fact that these mice still have DP thymocytes does not necessarily demonstrate that all earlier stages of T cell development are normal. Therefore, there is always the concern that effects seen at later stages of T cell development are a reflection of earlier alterations. This caveat should be discussed in the manuscript.

We thank the reviewer for this suggestion. One of the major findings of this work is indeed an antecedent role of Bcl11b in priming of lineage specifying genes, presumably before or at the transition to the DP stage. We indeed show that later stages of T cell development are influenced by earlier alterations by comparing the distinct phenotypes seen in germline Bcl11b hypomorphic mutant mice and mice in which Bcl11b is inactivated at the DP stage by Cd4-Cre. We stated this point in the second paragraph of the discussion section.

2. More importantly, the data shown throughout the manuscript lack all indications of absolute cell numbers. This information is critical to interpreting the results presented. For many of the experiments, the authors conclude that developing mature thymocytes are experiencing 'lineage-scrambling', implying that a cell which is receiving external signals to differentiate into one lineage is 'confused' and instead, is mis-expressing genes representative of both lineages. For this conclusion to be valid, the numbers of cells in the different genotypes of mice should be similar. Alternatively, if some genotypes of mice have very few thymocytes altogether, the flow cytometry data showing relative proportions of cells with various phenotypes would not really support the conclusions the authors are arriving at.

We thank the reviewer for raising this issue, which is related to point made by reviewer #2. We added data of absolute cell numbers in Figures (Figure S1e and Figure 4e) in the revised manuscript. Clearly numbers of mature thymocytes are significantly reduced by *Bcl11b* mutant mice, consistent with a previous report (Albu, D.I. et al. JEM 2007, 204:3003) showing an essential role of *Bcl11b* in positive selection. We agree with the reviewer that in the case where a molecule is solely involved in CD4/CD8 lineage choice (such as ThPOK), mature thymocytes numbers are not changed. This finding was interpreted to propose that positive selection and lineage choice are independent process (Keefer et al. Science 1999, 286:1149). However, this is not always the case and *Bcl11b* is also involved in positive selection, as was observed in Runx complexes (Setoguchi et al. Science 2008, 319:822). Therefore, our analyses in this study focused on how thymocytes selected through MHC-I and –II, even in an inefficient manner, differentiate into appropriate CD8⁺ cytotoxic and CD4⁺ helper lineages, by using MHC-II and MHC-I deficient mice, respectively. This is the most reliable way to address differentiation of MHC-I and –II selected cells. Please note that, in addition to the unusual proportion of cell subsets as defined by CD4/CD8 expression, we combined analyses of *Thpok* and *Runx3* expression, and provided genetic data showing that dys-regulated ThPOK expression is a main cause of CD4 skewed differentiation from *Bcl11b*^{m/m} progenitors (Figure 2a). We believe that these results sufficiently support our conclusion.

3. In many figures, certain panels of thymocytes are labeled ‘mature’. The figure legends do not clearly indicate what this means. Are all panels of ‘mature’ thymocytes gated on TCRβ⁺? What about Fig. 2E, which is not labeled ‘mature’ – are these gated on TCRβ⁺? If all analysis was done using TCRβ⁺ as the only criterion, what about other lineages of TCRβ⁺ thymocytes beyond conventional CD4 and CD8 T cells, such as NKT cells or other MHC class Ib-specific cells? This is likely only a major concern if total thymocyte numbers are vastly different between mice of different genotypes, but should be addressed at some level.

We thank the reviewer for this suggestion. We clarified the gating strategy to define ‘mature’ thymocytes in each experiment. We used a CD24⁻TCRβ⁺ gate to define mature thymocytes. Although this population contains NKT cells and FoxP3⁺ Treg cells, it was reported in reference #7 that *Bcl11b*^{F/F}:*Cd4-Cre* mice have a defect in NKT cell generation. We also showed that the FoxP3⁺ Treg subset is not generated from *Bcl11b*^{m/m} progenitors. Dysregulated expression of *Thpok* and *Runx3* from differentiated CD4⁺ T cells due to late *Bcl11b* inactivation (*Thpok-Cre* and *Retroviral Cre*) support the interpretation that the major phenotype we observed should reflect an impaired differentiation

pathway of conventional CD4 and CD8 lineage cells, although expansion of other MHC class Ib-specific cells is not formally excluded.

4. In general, the figure legends contain insufficient information. Given word limitations on manuscripts, this is somewhat understandable, but it is often difficult to decipher exactly what the data in a given figure are showing.

According to this suggestion, we have tried to provide sufficient information in figures and figure legends up to the word limitations.

5. The manuscript is not clear on which experiments are done with unmanipulated mice (not fetal liver chimeras) and which were done by reconstitution, and furthermore, whether any were done in neonatal versus adult mice. It seemed the authors stated that the *Bcl11b*^{m/m} mice died shortly after birth, yet these mice were used in many of the figures (Fig 2, 3, 4, 5). What were these mice - neonates? This information needs to be more clearly presented.

We thank the reviewer for these related criticisms. We modified the text and figure legends to increase clarity. Most of analyses addressing differentiation of *Bcl11b*^{m/m} cells were performed by reconstitution of T cell development in host mice. Data using neonates were the western blot in figure 2a, ChIP in Figure 2b and ATAC-seq in Figure 3e.

6. Some of the findings are less convincing than others. For example, the data from *Bcl11b*^{fl/fl} x CD4-cre MHC I-null mice shown in Figure 5D are not at all straightforward. Many of the cells are DN, rather than CD8⁺, as argued. Also, the reduced expression of Runx3-tdTom in CD8SP thymocytes transduced with ThPOK (Figure 6e) is also quite modest. Is this difference biologically significant?

We thank the reviewer for pointing out these issues. Since emergence of the DN population was not observed in control MHC I-null mice, this has biological significance in regard to impaired differentiation of MHC-II selected cells. But, we agree that a more strict definition of re-direction would be a differentiation into CD8⁺ T cells. We therefore revised the text, describing this as partial re-direction of MHC-II selected cells. Reduced expression of Runx3-tdTomato by ThPOK transduction in control CD8⁺ T cells was reproducible, although it was modest. This result suggests that the *Runx3* locus is less sensitive than the *Cd8* locus to repression by ThPOK transduction. It was reported that cytotoxic effector genes such as IFN- γ , whose expression in CD8⁺ T cells is known to require Runx3 activity (Cruz-Guilloty F et al, JEM 2009,206:51), was reduced by ThPOK transduction (Jenkinson SR et al,

JEN, 2007, 204:267). We therefore think Runx3 reduction by ThPOK transduction is biologically significant.

Reviewers' comments:

Reviewer #1.

The authors have addressed all my concerns in the revised manuscript. No further questions.

Reviewer #2.

The authors have made extremely useful responses to the reviewers' comments and have strengthened the clarity and power of this manuscript. It is an impressive and important contribution that reveals a great deal about how Bcl11b participates in different cell fate decisions, and along the way it sheds unexpected light on what the underlying logic of those decisions must be. For example, the difference in timing between the roles of Bcl11b in controlling Thpok and in controlling Runx3 expression is very interesting, as is the stark difference between the roles of Bcl11b C-terminal domains in ILC2 development, in DN cell commitment, and in regulation of Thpok and Runx3.

Small typos and minor issues that authors (or production editors) may wish to address:

--The new Fig. 4f should be mentioned in the text. One possible place is in line 248.

We mentioned Fig. 4F on page 12.

--On p. 13, regarding Fig. 5d, the reduction of Runx3-tdTomato by ThPOK seems weak even in control cells, limiting the dynamic range over which Bcl11b effects can be measured. Perhaps on line 275 it would be helpful to say "but not AS MUCH in Bcl11b-deficient cells", rather than the current wording which implies a more dramatic difference.

Accordingly, we described Runx3 repression as "Runx3-tdTomato was not repressed as much in CD4⁺CD8⁻ mature thymocytes" on page 13.

--In Fig. 3d, correct spelling of label to "DN3 thymocytes". Also, in Fig. 3e, shouldn't the y axes be RPM rather than RPKM for ATAC-seq?

We corrected these typos.

--In Fig. 5d, it would be helpful to add genotype labels to the CD4/CD8a flow cytometry panels.

We added genotype labels in the revised Figure 5d.

Reviewer #3.

The authors have satisfactorily addressed the concerns raised in the original reviews. The revised manuscript is substantially improved, and is recommended for publication.